# GRouNdGAN: GRN-guided simulation of single-cell RNA-seq data using causal generative adversarial networks

Yazdan Zinati ®[1], Abdulrahman Takiddeen ®[1] & Amin Emad ®[1,2,3] ✉

We introduce GRouNdGAN, a gene regulatory network (GRN)-guided reference-based causal implicit generative model for simulating single-cell RNA-seq data, in silico perturbation experiments, and benchmarking GRN inference methods. Through the imposition of a user-defined GRN in its architecture, GRouNdGAN simulates steady-state and transient-state single-cell datasets where genes are causally expressed under the control of their regulating transcription factors (TFs). Training on six experimental reference datasets, we show that our model captures non-linear TF-gene dependencies and preserves gene identities, cell trajectories, pseudo-time ordering, and technical and biological noise, with no user manipulation and only implicit parameterization. GRouNdGAN can synthesize cells under new conditions to perform in silico TF knockout experiments. Benchmarking various GRN inference algorithms reveals that GRouNdGAN effectively bridges the existing gap between simulated and biological data benchmarks of GRN inference algorithms, providing gold standard ground truth GRNs and realistic cells corresponding to the biological system of interest.

Unraveling gene regulatory interactions, often represented as a gene regulatory network (GRN), plays a crucial role in studying biological processes under different conditions[1–3], simulating knockdown and knockout experiments[4,5], and identifying therapeutic drug targets[6,7]. Many algorithms have been proposed for GRN reconstruction using bulk or single-cell RNA sequencing data (scRNA-seq), alone[8–14] or with other modalities[15–18]. While these advances have provided great biological insights, evaluating the performance of GRN Inference algorithms remains challenging[19,20] due to the lack of reliable ground truth for the biological processes under study. Existing evaluation approaches often resort to curated databases[21–23]. However, the regulatory interactions in these databases are aggregated from a wide range of datasets and are not specific to a biological system, making them not ideal benchmarks due to context specificity of gene regulation. Another strategy is to verify regulatory interactions by conducting perturbation experiments on the system under study[20]. However, this approach is tedious, lengthy, and expensive. Another approach is to employ scRNA-seq simulators. Despite great progress in this domain, most simulators lack the essential properties for this task, such as the preservation of gene identities and simulation based on a user-provided ground truth GRN. For example, scGAN[24], cscGAN[24], scDESIGN2[25], and SPARSIM[26] capture inter-gene correlations in their datasets. However, since they do not explicitly impose a known GRN (that can act as the ground truth), they are not suitable for benchmarking GRN inference methods.

A small subset of simulators (e.g., BoolODE[19], SERGIO[4], and GeneNetWeaver (GNW)[27]) explicitly incorporate GRNs capturing transcription factor (TF)-gene dynamics. GNW was used to generate benchmarks for Dialogue for Reverse Engineering Assessment and Methods (DREAM) challenges[27–29]. However, being a bulk RNA-seq tool, its simulated datasets do not replicate the distribution of experimental scRNA-seq datasets nor exhibit their statistical properties, despite attempts to adapt it for this purpose by externally inducing dropout events[10,30]. SERGIO and BoolODE were designed to simulate scRNA-seq

[1]Department of Electrical and Computer Engineering, McGill University, Montreal, QC, Canada. [2]Mila, Quebec AI Institute, Montreal, QC, Canada. [3]The Rosalind and Morris Goodman Cancer Institute, Montreal, QC, Canada. ✉e-mail: amin.emad@mcgill.ca

data using stochastic differential equations (SDEs)[4,19] and have been used to benchmark a variety of GRN inference methods. However, often a mismatch between the benchmarking results based on experimental and their simulated data have been reported, which may be attributed to differences between the simulated and experimental datasets. For example, in the BEELINE study (using BoolODE), some of the top ranking methods on simulated benchmarks reported near random performance on curated and experimental benchmarks[19]. Both simulators enable the user to simulate more realistic datasets by carefully selecting the values of the SDE parameters. Moreover, using a reference dataset, SERGIO allows fine tuning the added technical noise in an iterative procedure until the generated and reference datasets are matched[4]. While these steps can help improve the resemblance to real datasets, they are often suboptimal and pose an undue burden on the user: for example, in SERGIO, the user must evaluate the similarity based on five different statistics and change three parameters iteratively until desired resemblance is achieved (which itself can be subjective). In addition, since the GRN is imposed on the "clean" dataset, but technical noise is added afterwards, this step may change the encoded causal relationships in a non-trivial way (which may explain why the performance of GRN inference methods dropped close to random when applied to the noisy dataset[4]). Furthermore, SERGIO and BoolODE do not readily preserve gene identities and make simplifying assumptions regarding cooperative regulation (see Discussion). These shortcomings demonstrate the need for simulators capable of generating realistic scRNA-seq data that retain the regulatory dynamics specified by a user-defined GRN. Importantly, with the growing interest in causal inference, simulators that impose causal GRNs are of great need.

GRouNdGAN (GRN-guided in silico simulation of single-cell RNA-seq data using Causal generative adversarial networks) is a causal implicit generative model for reference-based GRN-guided simulation of scRNA-seq data inspired by CausalGAN[31]. Given an input GRN and a reference scRNA-seq dataset, it can be trained to generate simulated data that is both indistinguishable from the reference data and faithful to the causal regulatory interactions of the input GRN. Unlike model-based simulators that rely on simplifying assumptions regarding co-regulatory patterns, in GRouNdGAN these patterns are *learned* through complex functions. This allows it not to compromise on the underlying complexity of the system and to model elaborate regulatory dynamics. GRouNdGAN provides state-of-the-art performance in realistic scRNA-seq data generation, while preserving gene identities, causal gene regulatory interactions, and cellular dynamics (e.g., lineage trajectory and pseudo-time ordering). This is achieved through implicit parameterization without the need for manual fine-tuning. Using GRouNdGAN, we benchmark eight GRN inference methods and find the results to be aligned with BEELINE's experimental results[19]. Furthermore, the causal structure of GRouNdGAN enables it to be used for sampling from both interventional and observational data distributions, enabling in silico knockout experiments.

## Results

### GRouNdGAN generates scRNA-seq data using causal generative adversarial networks

GRouNdGAN is a deep learning model that generates scRNA-seq data while imposing a user-defined causal GRN to describe the regulatory relationships of the genes and TFs. Its architecture builds on the causal generative adversarial network[31] and includes a causal controller, target generators, a critic, a labeler and an anti-labeler (Fig. 1). Training includes two steps. First, the causal controller that generates TF expression values is pre-trained (Fig. 1B) as the generator of a Wasserstein GAN (WGAN)[32] with gradient penalty (WGAN-GP) (Methods). In the second step (Fig. 1C), TF expression values (generated by the pre-trained causal controller) and randomly generated noise are provided as input to the target generators that produce target genes' expressions, while incorporating the TF-gene relationships of the input GRN. To achieve this, as input, each target generator only accepts a noise value and the generated expression values of TFs that causally regulate it (Fig. 1C, Methods). The generated expression of genes and TFs are then fed to a library-size normalization (LSN) layer[24].

The two steps of training above are performed based on the same reference experimental scRNA-seq training dataset. The goal of the first step (pre-training, Fig. 1B) is to train the causal controller to learn scRNA-seq data distribution and generate realistic data, irrespective of any GRN. The GRN edges between a set of regulators (e.g., TFs) and their targets are imposed in the second step of training (Fig. 1C). To achieve this, in the second step of training, only the expression of regulators (TFs) outputted by the causal controller are used and its output of target genes' expressions are discarded and instead are re-generated using the target generator neural networks in Fig. 1C to enable imposition of causal relationships.

The critic's role (in both steps of training) is to quantify the Wasserstein distance (Supplementary Notes) between the reference and simulated data. The target generators are trained in an adversarial manner to generate realistic datapoints indistinguishable from reference datapoints by the critic. The labeler/anti-labeler estimate the TF expression values *only* from the target genes' expressions to ensure that the generated causal TF-gene dependencies are encoded. The combination of the model architecture (in that each target generator only receives a noise value and the simulated expression of its regulating TFs in the GRN) and the labeler/anti-labeler ensure that the GRN edges are causally imposed in the simulated data. Details are provided in Methods, the architectural choices in Supplementary Tables 1 and 2, and an ablation study in Supplementary Notes and Supplementary Table 3.

### GRouNdGAN generates realistic scRNA-seq data

We first trained GRouNdGAN on three datasets (results for three additional datasets will be discussed later). The first dataset contained scRNA-seq profiles of 68,579 human peripheral blood mononuclear cells (PBMCs) from 10x Genomics[33] corresponding to eleven cell types ("PBMC-All"). We formed a dataset of the most common cell type in the PBMC-All dataset containing 20773 CD8+ Cytotoxic T-cells ("PBMC-CTL"). Additionally, we obtained the scRNA-seq (MARS-seq) profile of 2730 cells corresponding to differentiation of hematopoietic stem cells to different lineages from mouse bone marrow[34] ("BoneMarrow"). We used GRNBoost2[8] to identify fifteen TFs for each gene to form the input GRN. Note that these identified interactions may contain both spurious correlations and causal relationships in the reference dataset. However, when they are used as input to GRouNdGAN, they will correspond to causal interactions in the *simulated* dataset (see Kocaoglu et al. for theoretical details of causality in this architecture[31]). For each dataset, we trained GRouNdGAN on a randomly selected training set and evaluated on a held-out test set (Methods, Supplementary Data 1 – Sheets 2 and 3). Figure 2A, B and Supplementary Figs. 1–3 show the t-SNE plots of reference and simulated cells, qualitatively revealing their similarity.

We quantitatively assessed the resemblance using Euclidean distance, Cosine distance, maximum mean discrepancy (MMD)[35], mean integration local inverse Simpson's index (miLISI)[36], and the area under the receiver operating characteristic curve (AUROC) of a random forests (RF) classifier distinguishing simulated and experimental cells (Methods). As a "control" (and to enable calibration of these scores), we also calculated these metrics using two halves of the reference test set (corresponding to real cells). Table 1 shows the performance of GRouNdGAN, control, and three state-of-the-art simulators for the PBMC-CTL dataset (see Supplementary Data 1 – Sheet 2 for training set performance). We did not include BoolODE in these tables since it is a reference-free simulator and is not designed to match a particular

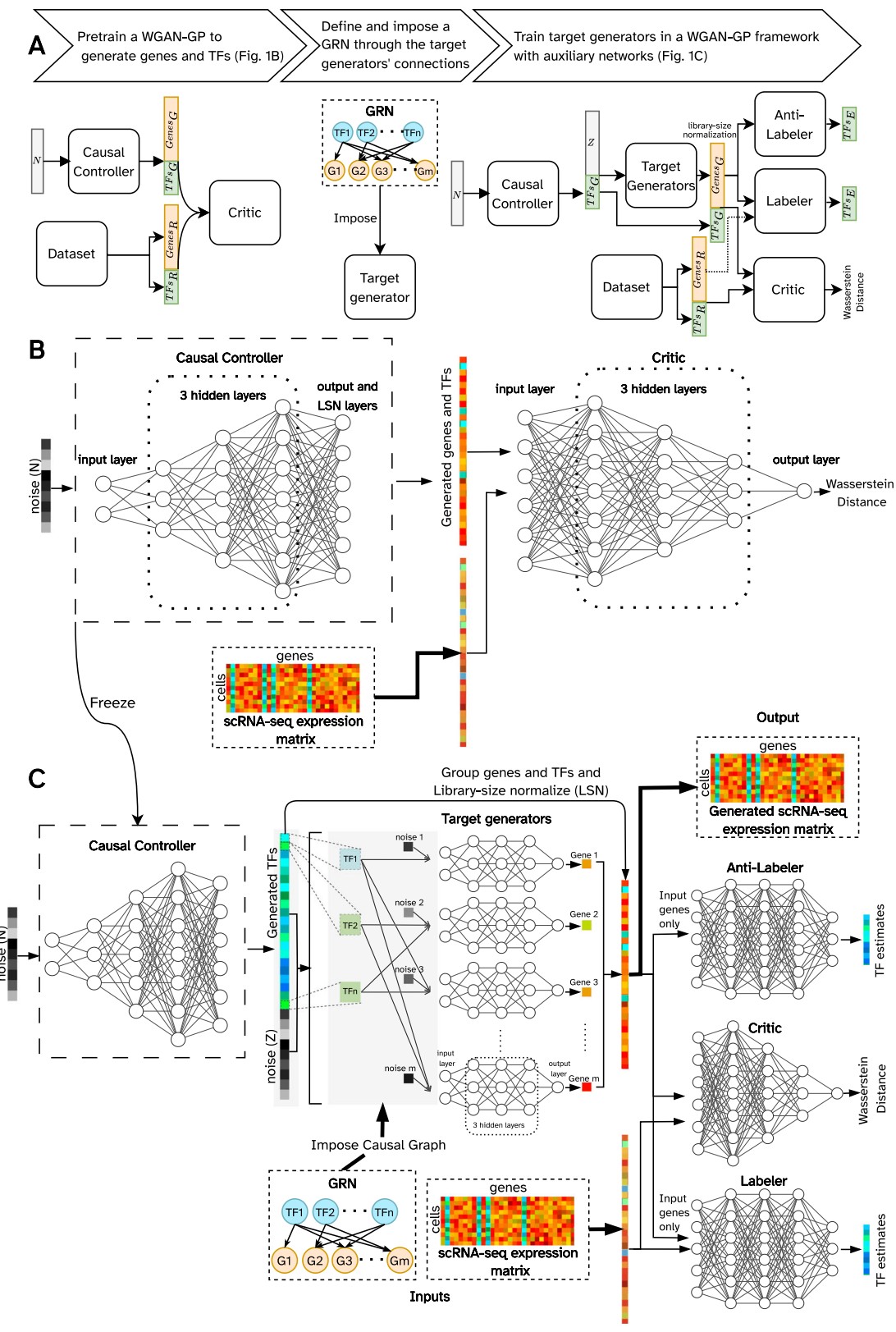

reference dataset. We tried generating samples using SERGIO to match the PBMC-CTL dataset, but the simulated dataset did not show high resemblance to the PBMC-CTL dataset (see Supplementary Notes, Supplementary Tables 4 and 5 and Supplementary Fig. 4). However, we decided not to include these results in Table 1, since SERGIO relies on the user's judgement for distribution matching and the quality of the final simulated dataset is user dependent. Instead, we only included

reference-based simulators that automatically generate scRNA-seq data resembling experimental data, in order to assess data quality systematically and fairly.

One of the challenges faced by GANs is the possibility of mode collapse: when applied to heterogenous datasets, they may learn to generate only a limited set of examples, instead of generating a variety of examples from the entire distribution of the training data. We

**Fig. 1 | Architecture and training procedure of GRouNdGAN. A** The flowchart representing the steps of the training procedure and the overall architecture of the model in each step. Subscripts G, R, and E represent generated, real, and estimated, respectively. **B** A WGAN-GP is pre-trained to generate realistic simulated cells using the reference (real) training set. **C** The LSN layer of the generator of the trained WGAN-GP (panel **B**) is removed, its weights are frozen, and is used as the causal controller to generate unnormalized TF expression values (expression of target genes generated by the causal controller are discarded). These TF expression values along with a noise vector are provided as input to the target generators, following the provided causal GRN. The generated gene and TF expression values

are reorganized and passed through the LSN layer. The normalized simulated expression vectors and experimental reference data (the same training set as **B** are then passed to the critic to estimate Wasserstein distance between the reference and the generated data distributions. The anti-labeler estimates TF values based on generated target gene expressions. The labeler performs a similar task, but in addition to receiving generated values, it also utilizes target gene expression values from the reference data. Labeler and anti-labeler ensure that the causal GRN is incorporated by the target generators. Details of the model are provided in Methods.

trained GRouNdGAN using PBMC-All, which contains multiple cell types, to determine whether it can generate samples from different cell types present in this dataset, or it will suffer from mode collapse and only generates samples corresponding to the most represented cell type. This is particularly important, since many scRNA-seq datasets are heterogenous and contain different cell types or cells in different states. If cell type information of the reference dataset is available a priori, one can easily generate realistic samples of each cell type separately (similar to the PBMC-CTL analysis) and use either cell type-specific GRNs, a shared GRN, or a combination of both. However, we were particularly interested to know whether our model could generate realistic samples *without* this knowledge, a more challenging task and useful for when such information is unavailable. Also, not requiring cell type information reduces the amount of user involvement (to annotate cell types) and improves model's usability. We repeated the analysis using the PBMC-All dataset, containing eleven cell types. Although GRouNdGAN did not receive cell type/cluster information, we did not observe mode collapse (Supplementary Fig. 2), and it was able to generate cells from distinct clusters (Fig. 2B, test set miLISI = 1.90). GRouNdGAN outperformed all simulators that did not use cell cluster information (Table 2) and compared to those that utilized this information, it was still the top performing based on MMD and RF AUROC and the second best according to the Euclidean distance (6% higher than the best) and miLISI (0.5% lower than the best) (Table 2). Moreover, we observed a high degree of concordance between the cell type marker expressions in the experimental data and GRouNdGAN-simulated data (Supplementary Notes and Supplementary Fig. 5). Finally, GRouNdGAN outperformed all other simulators based on all metrics when we repeated the analysis using the BoneMarrow dataset, containing continuous cell states and much fewer cells (Supplementary Data 1 – Sheet 3). These results show that GRouNdGAN is a stable simulator of scRNA-seq data and even without extra cell type information in heterogenous datasets, it automatically generates realistic samples reflecting different cell types (while imposing a GRN). See Supplementary Notes and Supplementary Data 2 for stability analysis and the effect of different GRN properties (including number of TFs) on the performance.

## GRouNdGAN imposes a causal GRN in the simulated data
To assess the ability of GRouNdGAN in imposing the input causal GRN, we performed in silico TF knockout experiments (one of GRouNdGAN's capabilities) on the simulated cells ($n = 1000$) from the PBMC-CTL dataset. We performed a forward pass after setting the expression of each TF to zero (one at a time) in the causal controller output, while keeping all other parameters and TF expressions unchanged, ensuring that the perturbations were performed on the same batch of cells forming matched case/control experiments. There was no change in the expression of genes that were not regulated by the knocked-out TF (as expected), while the expression of the regulated genes changed. Figure 2F shows the distribution of the adjusted *p*-values (Benjamini–Hochberg) for each TF-gene edge in the GRN, comparing the expression of a gene across all cells before and after the knockout of one of its regulating TFs (two-sided Wilcoxon signed-rank tests). In the majority (66.5%) of cases, the knockout of a gene's

regulating TF significantly (adjusted *p*-value < 0.05) altered its expression, showcasing that the GRN is indeed imposed in the expression profiles of simulated cells. Note that since each gene is regulated by multiple TFs, we cannot expect that knockout of a single TF result in a significant change in the expression of its target genes in all cases. The analysis above shows that GRouNdGAN does not simply ignore the input TF values to instead rely solely on the noise input to generate realistic cells, and the effect of TF-gene edges are indeed imposed. Moreover, since each gene is regulated by multiple (15) TFs, but the knockout experiment is one TF at a time, this further shows that individual edges are imposed by the model.

GRouNdGAN does not require the input GRN to be the true causal GRN of the training data (we will discuss this further later). To further clarify this, we replaced GRNBoost2 with PPCOR[37] to form the GRN, which is a method based on (partial) correlation for inferring GRNs (see Supplementary Data 1 – Sheet 2 for resemblance metrics). Repeating the knockout experiments above ($n = 1000$ cells) resulted in similar results showing that in the majority (75.2%) of cases, the knockout of a gene's regulating TF significantly altered its expression (Supplementary Fig. 6). Taken together, these results show that GRouNdGAN can successfully impose different GRNs in the simulated data, while generating realistic cells.

## The imposed GRN can be reconstructed from the simulated data
To test whether the imposed GRN can be reconstructed from the generated data, we first applied GRNBoost2[8] to the reference PBMC-CTL dataset and recorded the top ten TFs for each gene. For each gene, we created two sets of five TFs based on their even/odd parity in the ranked list: TFs ranked 1st, 3rd, 5th, 7th, and 9th were connected to the gene and were imposed ("positive control GRN"), while the remaining five TFs were not ("negative control GRN"), resulting in two GRNs with identical densities and the same number of TFs and target genes. When considering the positive and negative control GRNs as ground truths (separately), the GRN reconstructed using GRNBoost2 from the reference (training) dataset ($n = 19,773$) resulted in comparable AUPRC (0.28 and 0.24, respectively, Supplementary Data 3 – Sheet 2), reflecting that they contain edges of comparable importance.

We simulated data ($n = 19,773$ cells) using GRouNdGAN by imposing only the positive control GRN, and used GRNBoost2, PIDC[10] GENIE3[9], and PPCOR[37] to reconstruct the underlying GRN (Fig. 3, Supplementary Fig. 7, Supplementary Data 3 – Sheet 2). All methods reconstructed the imposed edges with a much higher AUPRC compared to the reference dataset, while they did not assign high scores to the unimposed edges, resulting in close to random AUPRC. Figure 3 shows the performance of GRNBoost2 applied to the reference, GRouNdGAN-simulated, and scGAN-simulated data ($n = 19,773$ cells). The performance for scGAN (both GRNs) was lower than the reference dataset, showcasing that it had not preserved the TF-gene relationships of the experimental dataset well. The performance was relatively consistent between positive (Fig. 3A, C) and negative control (Fig. 3B, D) GRNs using the reference and scGAN-simulated data; however, it was much higher for the positive control GRN (compared to negative control GRN) using GRouNdGAN-simulated data. Similar patterns were observed when we repeated these analyses using the

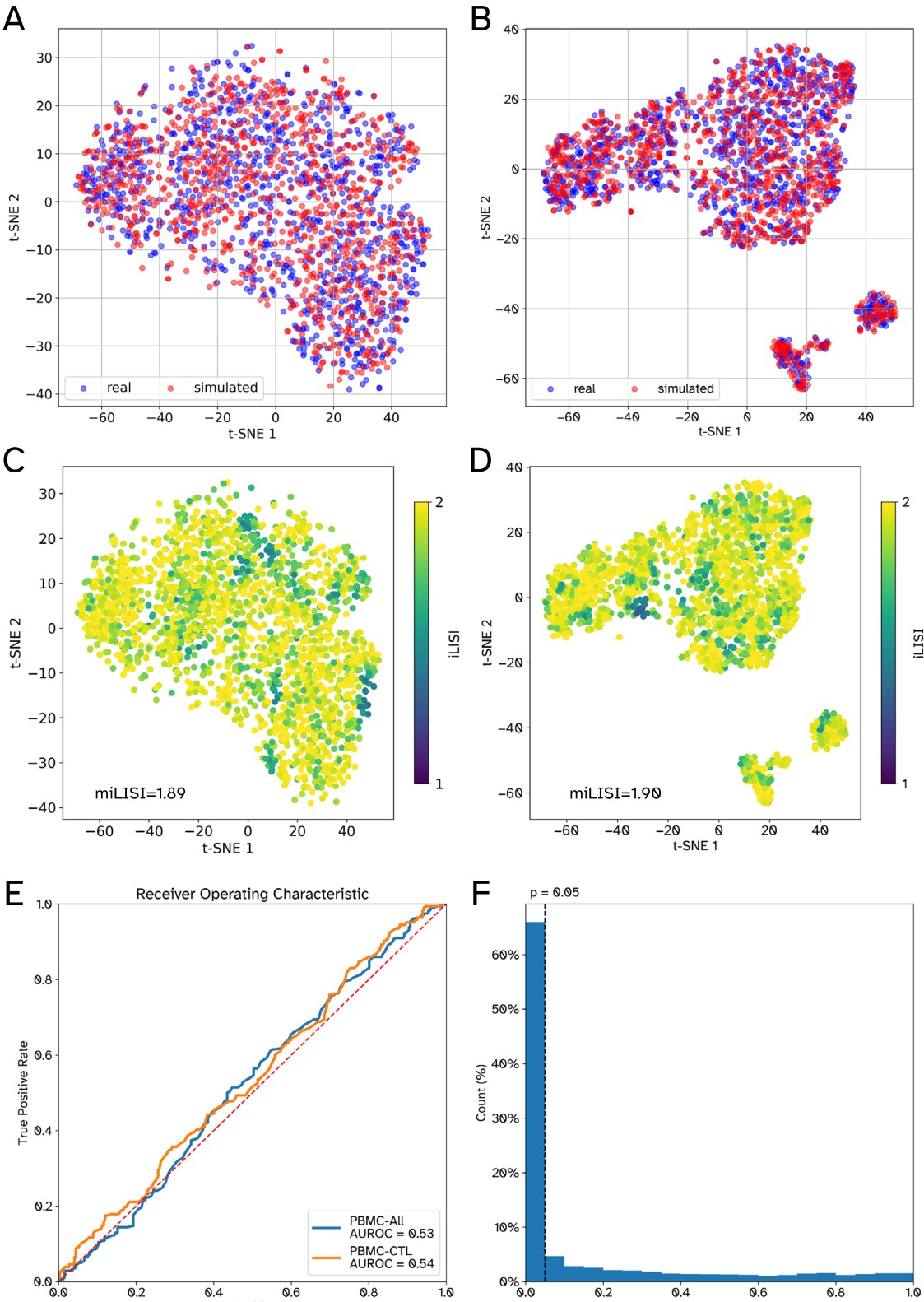

**Fig. 2 | Performance of GRouNdGAN in generating realistic scRNA-seq data.** All plots correspond to 1000 real (experimental) held-out test cells and 1000 simulated cells. Each gene in the imposed GRN of GRouNdGAN is regulated by 15 TFs (identified using GRNBoost2 from the experimental training set). Panels **A** and **B** depict the t-SNE plots of real (blue) and simulated (red) cells for PBMC-CTL and PBMC-All, respectively, while panels **C** and **D** also show their iLISI distributions. **E** The ROC curve of a Random Forest classifier in distinguishing between simulated and real cells. **F** The distribution of adjusted *p*-values corresponding to the TF perturbation study on the PBMC-CTL dataset. The *p*-values are obtained using two-sided Wilcoxon signed ranked tests and are adjusted for multiple hypotheses following the Benjamini–Hochberg procedure. Source data are provided as a Source Data file.

**Table 1 | Performance of different simulators in generating realistic scRNA-seq data using the PBMC-CTL dataset**

| Simulator | Cosine distance | Euclidean distance | MMD | RF AUROC | miLISI |
|---|---|---|---|---|---|
| GRouNdGAN | **0.00057** | **182** | **0.026** | **0.54** | **1.891** |
| scGAN[24] | 0.00095 | 222 | 0.031 | 0.59 | 1.888 |
| scDESIGN2[25] | 0.00100 | 229 | 0.065 | 0.76 | 1.736 |
| SPARsim[26] | 0.00104 | 235 | 0.309 | 0.95 | 1.625 |
| Control | 0.00019 | 99 | 0.012 | 0.50 | 1.909 |

The metrics are calculated between a simulated dataset of 1000 cells and the held-out test set of 1000 real cells (see Supplementary Data 1 – Sheet 2 for training set performance). Each gene in the imposed GRN of GRouNdGAN is regulated by 15 TFs (constructed using GRNBoost2 from the experimental training set). For the first three metrics, a value closer to zero is preferred, for RF AUROC a value closer to 0.5 is preferred, and for miLISI a value closer to 2 is preferred. For the first two metrics, the values correspond to the distance of the mean centroids of the real and simulated cells. The RF AUROC of control corresponds to perfect performance (of a random classifier). The other control metrics are calculated using the two halves of the real test dataset. Best performance values (excluding control) are in bold-face.

**Table 2 | Performance of different simulators in generating realistic scRNA-seq data using the PBMC-All dataset**

| Simulator | Cell cluster labels/ ratios provided as input | Cosine distance | Euclidean distance | MMD | RF AUROC | miLISI |
|---|---|---|---|---|---|---|
| GRouNdGAN | No | 0.00028 | 168 | **0.017** | **0.53** | 1.90 |
| cscGAN[24] | Yes | 0.00025 | 242 | 0.030 | 0.56 | **1.91** |
| cWGAN | Yes | 0.00053 | 239 | 0.027 | 0.57 | 1.89 |
| scGAN[24] | No | 0.00081 | 300 | 0.035 | 0.58 | 1.90 |
| scDESIGN2[25] | Yes | 0.00032 | 181 | 0.046 | 0.86 | 1.81 |
| scDESIGN2[25] | No | 0.00057 | 256 | 0.124 | 0.91 | 1.53 |
| SPARsim[26] | Yes | **0.00019** | **158** | 0.287 | 0.96 | 1.66 |
| SPARsim[26] | No | 0.00035 | 245 | 0.307 | 0.96 | 1.39 |
| Positive control | NA | 0.00022 | 116 | 0.012 | 0.50 | 1.91 |

Baseline models that enable utilization of cell cluster labels or ratios are run with and without this information. The metrics are calculated between a simulated dataset of 1000 cells and the held-out test set of 1000 real cells. Each gene in the imposed GRN of GRouNdGAN is regulated by 15 TFs (constructed using GRNBoost2 from the experimental training set). For the first three metrics, a value closer to zero is preferred, for RF AUROC a value closer to 0.5 is preferred, and for miLISI a value closer to 2 is preferred. For the first two metrics, the values correspond to the distance of the mean centroids of the real and simulated cells. The RF AUROC of control corresponds to perfect performance (of a random classifier). The other control metrics are calculated using the two halves of the real test dataset. Best performance values (excluding control) are in bold-face.

PBMC-All ($n = 67,579$ cells) and BoneMarrow ($n = 2230$ cells) datasets (Supplementary Data 3 – Sheets 3 and 4, Supplementary Figs. 8 and 9). These results show that the imposed edges were accentuated by GRouNdGAN, while the unimposed edges were disrupted and could not be found by GRN inference methods.

Next, we asked whether GRNBoost2 can reconstruct the imposed GRN, if GRNs of different densities are used to simulate data. Using data for which each gene was regulated by 15, 10, 5, and 3 TFs, we observed that in all cases the GRN imposed by GRouNdGAN could be inferred from the simulated data (Supplementary Data 3 – Sheets 5 and 6). Finally, we repeated the controlled analysis above, swapping the role of positive and negative control GRNs. Similar to the results of Fig. 3, the imposed edges were inferred by GRNBoost2, while the unimposed edges were not accurately discovered (Supplementary Data 3 – Sheet 7). These results show that 1) GRouNdGAN imposes the causal GRN, 2) the GRN inference methods can identify the imposed edges, and 3) the TF-gene relationships present in the reference

dataset but unimposed by GRouNdGAN are disrupted during simulation. The last property is particularly desirable for benchmarking GRN inference methods to ensure that the regulatory relationships present in the reference dataset (but unimposed) do not bias the simulated data, which could result in an inflated false positive rate.

Next, we repeated the analyses above on the PBMC-CTL dataset, but this time with input GRNs learned by PPCOR (a GRN inference based on partial correlation) from the training set. Similar to the results above, GRNBoost2 could reconstruct the imposed edges, and the importance of the imposed edges were accentuated by GRouNdGAN (compared to the original training dataset) (Supplementary Data 3 – Sheet 8). We also imposed two randomly generated GRNs (with similar density to the GRNs above in which each gene was regulated by 5 TFs). Once again, the imposed edges were accentuated by GRouNdGAN, however imposing a random GRN (whose induced patterns may be inconsistent with real reference data) came at the cost of lower resemblance between simulated and reference data. (See Supplementary Notes and the Discussion for more details on why imposing a GRN that is inconsistent with the reference data (e.g., a random GRN) is a contradictory requirement to generating simulated data resembling the reference data).

## GRouNdGAN-simulated data preserves trajectories and can be used for pseudo-time inference

In addition to deciphering discrete states, scRNA-seq data is often used to determine continuous cell transitions during biological processes such as differentiation, using trajectory and pseudo-time inference[38-44]. We set to determine whether GRouNdGAN-simulated data conforms to the transitional states and pseudo-time of the reference data. This is particularly important for benchmarking GRN inference methods that utilize pseudo-time information[12-14]. For this purpose, we used the BoneMarrow dataset corresponding to differentiation of hematopoietic stem cells to different lineages and simulated $n = 2230$ cells using GRouNdGAN (same as the number of real cells in the reference training dataset, to remain consistent).

We used Partition-based graph abstraction (PAGA)[38] for trajectory inference of the reference and GRouNdGAN-simulated data. PAGA computes graph-like maps of data manifolds faithful to the topology of data, retaining its continuous and discrete structures. Supplementary Fig. 10 shows the PAGA graphs, where nodes capture discrete states and the edges signify transitions between them. To compare the identified trajectories, we used markers of the cell types present in this dataset (Supplementary Table 6)[38]. Figure 4 shows the expression patterns of two marker genes for erythroid cells (*Gata1, Klf1*), neutrophils (*Mpo, Ctsg*), and basophils (*Mcpt8, Prss34*) for PAGA trajectory graphs of both datasets (Supplementary Figs. 11–17 show marker genes of all cell types). The comparison showed similar activation patterns between corresponding regions of PAGA graphs of the experimental and GRouNdGAN data, revealing several trajectories (e.g., erythroid and neutrophil branches). Strikingly, even the number of nodes activated in each figure scaled similarly between the two datasets: e.g., the markers of basophils were consistently expressed in only one node in each dataset (Fig. 4E, F).

We annotated different cell types present in both datasets using these marker genes (following PAGA's official tutorial) (Fig. 5A, B). The GRouNdGAN's annotated graph showed similar topological features to that of the reference data and captured known biological properties of hematopoiesis, which were also noted by Wolf et al.[38] using the reference dataset (e.g., proximity between monocytes and neutrophils or association between megakaryocyte and erythroid progenitors). These observations show that the key characteristics of the reference data are retained in GRouNdGAN-simulated data.

We used the diffusion pseudotime algorithm[39] (in scanpy[45]) to infer the progression of cells through geodesic distance (Fig. 5C, D). We noticed strong concordance when examining pseudo-time values

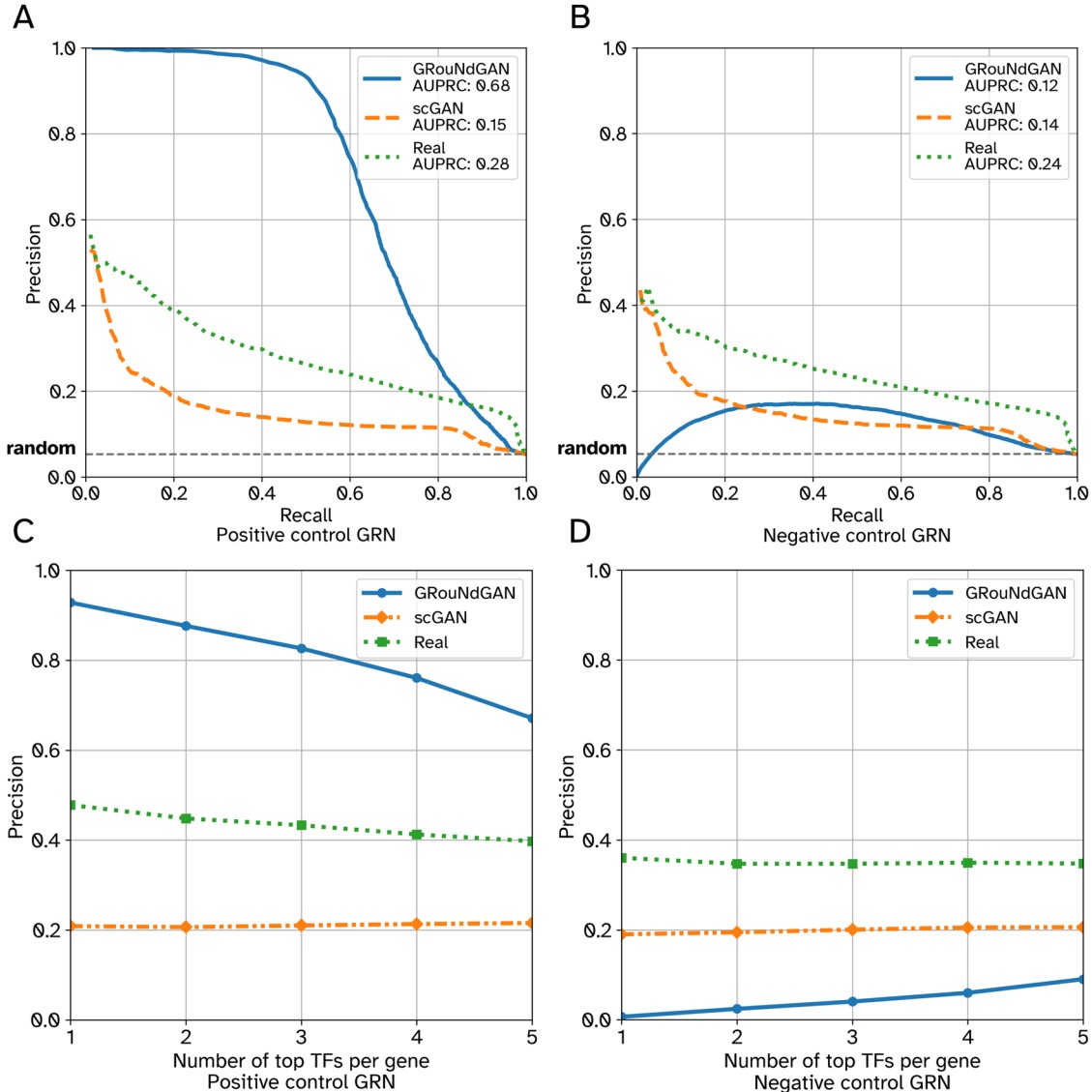

**Fig. 3 | Performance of GRNBoost2 in recovering the imposed edges versus unimposed edges (PBMC-CTL).** Left column shows the AUPRC (**A**) and Precision at k (per gene) (**C**) when the imposed edges (positive control GRN) were considered the ground truth. Right column shows the AUPRC (**B**) and Precision at k (per gene) (**D**) when the unimposed edges (negative control GRN) were considered the ground truth. Precision at k (per gene) refers to the precision when top k TFs for each gene is used to form the reconstructed GRN. All three datasets had the same number of cells ($n = 19,773$). Source data are provided as a Source Data file.

assigned to cell subgroups in the reference and GRouNdGAN-generated data. To track these patterns, we focused on marker expression changes along erythroid, neutrophil, and monocyte trajectories (Supplementary Figs. 11–13), similar to the analysis performed by Wolf et al.[38]. In both datasets, the activation of neutrophils' marker, *Elane*, and monocytes' marker, *Irf8*, were predominantly observed towards the later stages of their trajectories (Fig. 5E, F). Additionally, we observed the activation of erythroid maturity marker genes *Gata2*, *Gata1*, *Klf1*, and *Hba-a2* roughly in sequential order along the erythroid trajectory in both datasets (Fig. 5E, F), concordant with previous findings[38]. The striking similarity between the activation patterns of these markers in both datasets and the preservation of their ordering highlights GRouNdGAN's ability to capture dynamic transcriptional properties of scRNA-seq data, leading to correct trajectory inference and pseudo-time ordering.

### GRouNdGAN achieves high performance on other datasets

In addition to the datasets discussed earlier, we applied GRouNdGAN to three other datasets. One corresponded to cells undergoing hematopoiesis (similar to the BoneMarrow dataset), but with much larger number of cells ($n = 44,802$)[46] (which we called Dahlin dataset). The second dataset corresponded to malignant cells and cells present in the tumor microenvironment of 20 fresh biopsies from follicular lymphoma tumors ($n = 136,147$), which we called the Tumor-All dataset[47]. We also formed a dataset containing only the malignant cells present in this dataset ($n = 89,203$ cells), which we called the Tumor-malignant dataset. Table 3 and Supplementary Data 1 - Sheet 2 show the performance of GRouNdGAN in generating realistic scRNA-seq data, while Supplementary Figs. 18–20 show the tSNE plots of simulated and real cells. For all three datasets, different metrics show that the generated data has a high degree of similarity to experimental data and there is a small difference between the training and test set performances.

We next repeated the trajectory inference, pseudo-time ordering, and analysis of cell type marker expression along the pseudo-time axis on the Dahlin dataset using both GRouNdGAN-simulated data ($n = 43,802$ cells) and real cells ($n = 43,802$), as we had done using the BoneMarrow dataset (Supplementary Fig. 21). Similar to the results

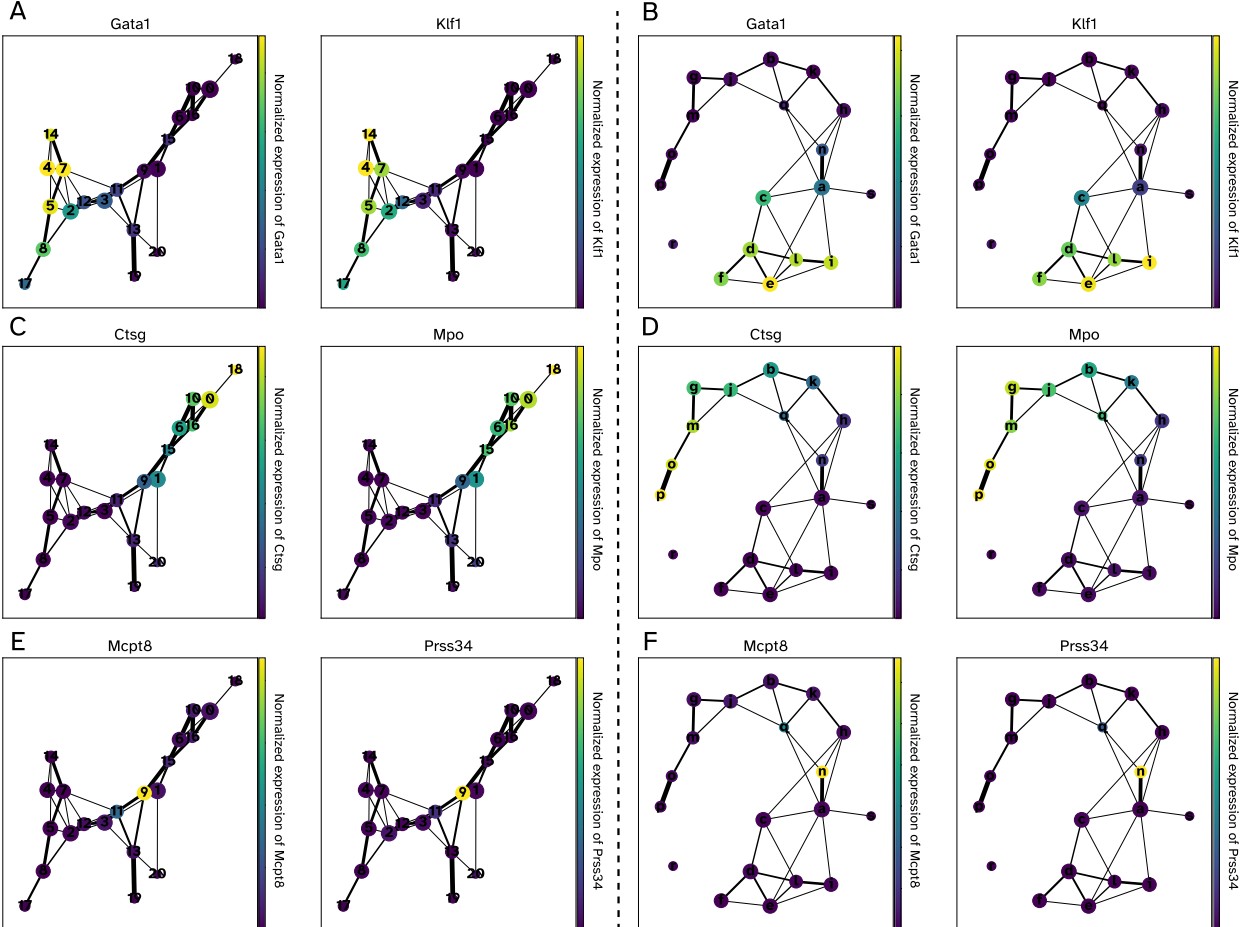

**Fig. 4 | PAGA graphs and cell type marker gene activation patterns in experimental and simulated BoneMarrow dataset.** In each figure, nodes represent Louvain clusters (capturing discrete states), edges represent transitions between states, and edge weights indicate confidence in the presence of connections. Low-connectivity edges below a threshold of 0.01 were discarded. Left panels (**A**, **C**, **E**) show the PAGA graph computed on GRouNdGAN-generated cells, and right panels (**B**, **D**, **F**) show the PAGA graph for the original BoneMarrow dataset. Node colors represent normalized expression of marker genes for erythroid cells (**A**, **B**), neutrophils (**C**, **D**), and basophils (**E**, **F**). Source data are provided as a Source Data file.

from the BoneMarrow dataset, we observed a high degree of concordance between the activation patterns of cell type markers in the simulated and real datasets and the preservation of their orderings. Taken together, these results suggest that GRouNdGAN can be used to generate realistic data from different biological contexts, while imposing a causal GRN and preserving various biological properties.

**Benchmarking GRN inference methods using GRouNdGAN confirms prior insights from curated and experimental datasets**
The BEELINE study[19] found conflicting results when benchmarking GRN inference methods on curated benchmarks and synthetic ones: top two performing models on synthetic data where among the bottom three on curated benchmarks. We re-investigated this using GRouNdGAN-simulated data with the BoneMarrow dataset and the positive and negative control GRNs (described earlier). We used seven GRN inference methods used by BEELINE, GRNBoost2[8], GENIE3[9], PIDC[10], PPCOR[37], LEAP[12], SCODE[13], and SINCERITIES[14], capturing a wide range of methods with a broad spread of performances reported on synthetic and curated benchmarks in the BEELINE study. We also included CeSpGRN[48], which is a cell-specific GRN inference method. We observed a striking resemblance between the performance pattern (order) of the methods on GRouNdGAN-simulated data and curated benchmarks from BEELINE (Fig. 5G, H and Supplementary Data 3 – Sheet 3 versus Fig. 4 of BEELINE[19]): the only difference was the swapping of LEAP and SCODE's order. We repeated this analysis using the

PBMC-CTL dataset (Supplementary Data 3 – Sheet 2), obtaining similar patterns. We also tested these methods using the PBMC-All dataset (Supplementary Data 3 – Sheet 4). While the results were generally consistent, the order of GENIE3 and GRNBoost2 were swapped, which may be due to their ability in working with datasets containing multiple distinct cell types (also see Discussion).

Overall, PIDC[10] outperformed all methods using both datasets, followed by GENIE3[9] and GRNBoost2[8]. LEAP[12], SCODE[13] and SINCERITIES[14] (which use pseudo-time and could only be applied to the BoneMarrow dataset) performed worse than others, matching their behavior in the BEELINE study on curated benchmarks (but not on synthetic benchmarks). All methods performed close to random on unimposed negative control GRN edges. These results not only show that GRouNdGAN can be used for benchmarking GRN inference methods, but also shows that the insights obtained from it matches those obtained from curated and experimental benchmarks (such as those used in Figs. 4–6 of the BEELINE study), the formation of which requires extensive amount of resources and effort.

**In silico perturbation experiments using GRouNdGAN**
In many studies, one is interested to characterize the relationship between TFs' expression and phenotypic labels (e.g., cell types). Differential expression (DE) analysis is a common approach to identify TFs (or other genes) most associated with a specific cell type. However, to directly test whether perturbing the expression of such candidate

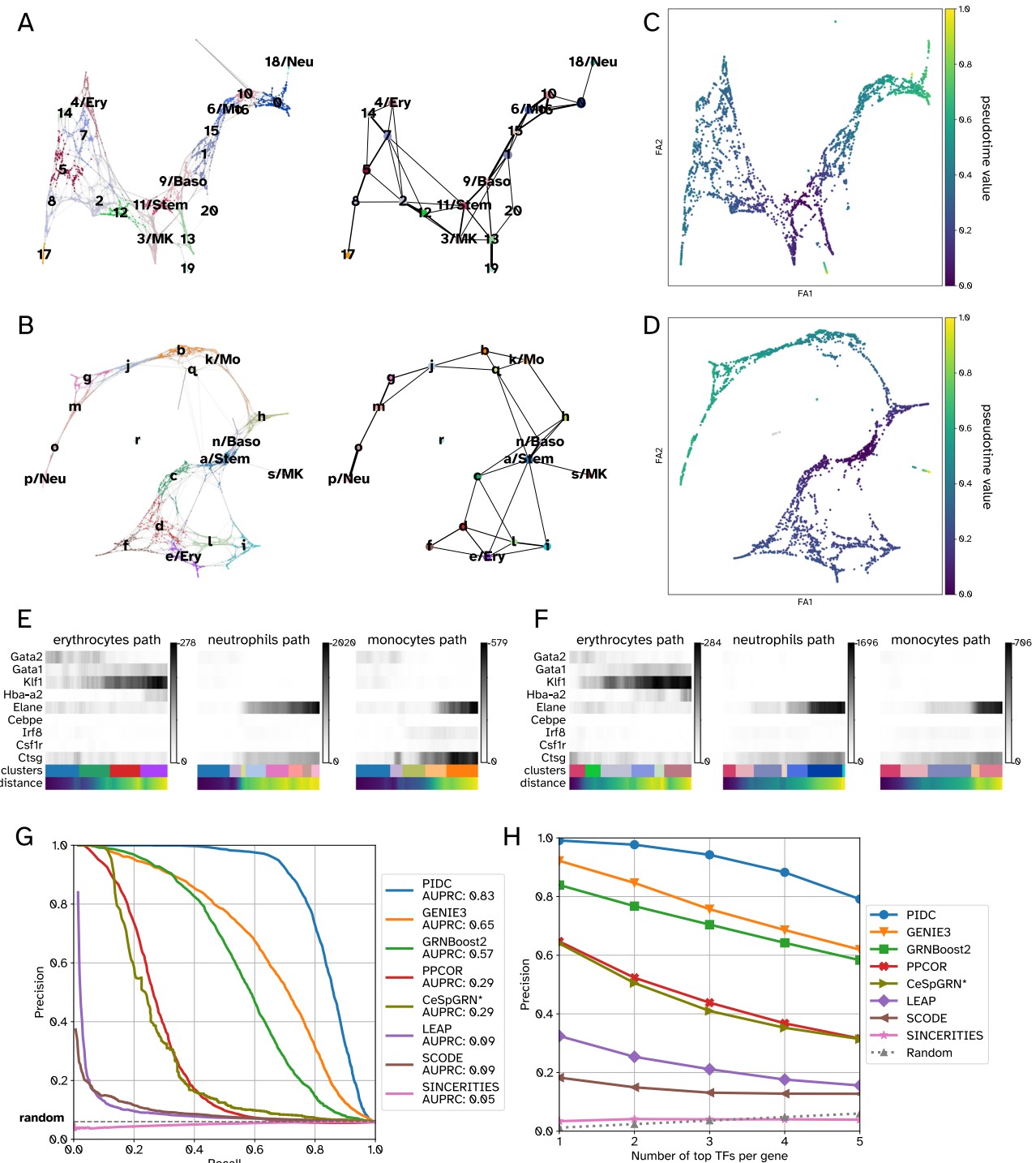

**Fig. 5 | Consistency in trajectories and pseudo-temporal orderings between data generated by GRouNdGAN and the experimental BoneMarrow hematopoietic dataset.** Panels **A** and **B** show the cell type annotations of simulated and experimental data, respectively. The figures on the left show the force-directed graphs (obtained using ForceAtlas2[66]) and the figures on the right show the PAGA graphs. Panels **C** and **D** show the PAGA-initialized force-directed graphs with cells colored by their inferred pseudo-time value. The following abbreviations are used: Stem for stem cells, Ery for erythroid cells, Neu for neutrophils, Mo for monocytes, MK for Megakaryocytes, and Baso for Basophils. Panels **E** and **F** display expression changes in the marker genes of erythrocyte (*Gata2*, *Gata1*, *Klf1*, and *Hba-a2*),

neutrophil (*Elane* and *Cebpe*), and monocyte (*Irf8*, *Csf1r*, and *Ctsg*) branches for experimental (**E**) and simulated data (**F**). The distance refers to the geodesic distance from a root cell (here a stem cell). A darker shade of gray shows higher expression. Panels **G** and **H** show the performance of different GRN inference methods on data generated by GRouNdGAN using the BoneMarrow dataset. The benchmarking results are based on a dataset with $n = 2230$ simulated cells (except for CeSpGRN). Source data are provided as a Source Data file. * Due to the high memory requirement of CeSpGRN, we were only able to run it on a subset of data containing $n = 1000$ cells and $m = 100$ genes (the set of regulating TFs and the edges connecting them to these genes remained unchanged).

**Table 3 | Performance of GRouNdGAN in generating realistic scRNA-seq data using the Dahlin, Tumor-All, and Tumor-malignant datasets**

| | Dahlin | | Tumor-All | | Tumor-malignant | |
|---|---|---|---|---|---|---|
| | Training | Testing | Training | Testing | Training | Testing |
| Cosine distance | 0.00011 | 0.00012 | 0.00141 | 0.00061 | 0.00059 | 0.00098 |
| Euclidean distance | 51 | 53 | 227 | 145 | 134 | 174 |
| MMD | 0.014 | 0.026 | 0.020 | 0.021 | 0.014 | 0.015 |
| RF AUROC | 0.52 | 0.54 | 0.53 | 0.52 | 0.53 | 0.57 |
| miLISI | 1.91 | 1.91 | 1.90 | 1.89 | 1.90 | 1.89 |

Each gene in the imposed GRN of GRouNdGAN is regulated by 15 TFs (constructed using GRNBoost2 from the experimental training set). For the first three metrics, a value closer to zero is preferred, for RF AUROC a value closer to 0.5 is preferred, and for miLISI a value closer to 2 is preferred. The metrics are calculated between a simulated dataset of 1000 cells and a set of 1000 real cells (all cells in the test set and 1000 randomly selected cells from the training set). For the first two metrics, the values correspond to the distance of the mean centroids of the real and simulated cells.

TFs change the expression of genes representative of a cell type, perturbation experiments (e.g., TF knockout) are desired. As a causal implicit generative model, GRouNdGAN provides the capability to sample from interventional distributions, thereby enabling in silico perturbation experiments. This functionality is provided at inference time by enabling manipulation of TFs' expressions as the outputs of the causal controller. This change is propagated along the GRN (through the model architecture), impacting the expression of genes regulated by the perturbed TF(s). By enabling a deterministic mode of operation, GRouNdGAN allows for exact comparison of gene expressions before and after perturbation while maintaining the invariance of other parameters (Methods).

To test this functionality, we asked whether the knockout of top 3 TFs most differentially expressed (Mann Whitney U test) between each cell type and the rest (identified from the reference dataset) would result in less cells being generated in the vicinity of the cells of that type. If positive, this implies that the perturbation in the TFs' expression has modified the gene expression profile of generated cells so that they no longer resemble that particular cell type (confirming our expectation). Figure 6A shows the UMAP embedding of the reference dataset (miLISI = 1.94 when compared with simulated data) and Fig. 6B shows the density plots of the experimental and simulated data, revealing a high degree of resemblance. We focused on the changes in the iLISI values of experimentally profiled cells of a specific type (annotated in the original study), when jointly mapped to the same embedding space with simulated data, before and after knockout. For the changes in the iLISI values to be observable, it was necessary to focus on cell types that occupy relatively distinct regions of the embedding space: CD14+ monocytes, CD19+ B cells, Dendritic cells, CD56+ natural killer cells (Supplementary Fig. 22); otherwise, the presence of other cell types in that region would confound the results.

Figure 6C shows the iLISI distributions for each cell type (only for real cells), calculated along with unperturbed simulated cells (blue) and perturbed simulated cells (orange). Figure 6D, E show the iLISI values and the distribution of generated cells after the knockout experiment for CD19+ B cells as an example (see Supplementary Figs. 23–26 for other cell types). In all cases, the iLISI values significantly reduced after the knockout experiment (Fig. 6C, Supplementary Table 7), showcasing that after the knockout of top TFs associated with a cell type, GRouNdGAN generated much fewer cells resembling that cell type (even though it did not have the knowledge of the cell type annotations). Instead, the simulated cells were dispersed into other cell types, yet they retained meaningful positions within the overall dataset embedding space. Also, the miLISI

of the other cell types remained relatively unchanged (Supplementary Table 7).

Next, we repeated the analyses above using Tumor-All dataset that contained different cell types including malignant cells, but also other cell types present in the tumor microenvironment. We considered cell types that made up of at least 0.25% of the dataset (Supplementary Table 8). When cells of two or more types populated the same regions of the embedding space, we kept the cell type containing more cells. Malignant cells, T cells, B cells, and plasma cells satisfied the conditions above, which we included in our analyses. Similar to the previous analyses, we observed that the KO of top 3 TFs of each cell type results in a significant reduction in the iLISI values (Supplementary Fig. 27, Supplementary Table 8).

These analyses suggest that in silico TF perturbation experiments using GRouNdGAN produces results concordant with results directly obtained from real biological data, a direction that we will explore further in the future studies.

## Discussion

GRouNdGAN is a causal implicit generative model designed to simulate realistic scRNA-seq datasets based on a user-defined GRN and a reference dataset. By incorporating GRN connections in its architecture and including auxiliary tasks, it imposes causal TF-gene relationships in the gene expression profile of simulated cells. These causal relationships are verifiable by in silico knockout experiments and identifiable by GRN inference methods. (In addition to the experimental analyses performed in this study, an interested reader should refer to Kocaoglu et al.[31] for theoretical evidence of causality of data generated by this architecture.)

We demonstrated that GRouNdGAN achieves state-of-the-art performance on various tasks including realistic scRNA-seq data generation, GRN inference benchmarking, and in silico knockout experiments. Even when applied to heterogenous datasets, it generated realistic datasets without utilizing information regarding cell clusters or cell types, achieving best or second-best performance compared to models that used this information. We should note that it is trivially possible to use cell type/cluster information with GRouNdGAN: one can simply provide subsets of the experimental dataset corresponding to a cell type/cluster as a reference, one at a time, with a cell-type specific (or shared) GRN (similar to our analyses using PBMC-CTL and PBMC-NaiveT in Supplementary Data 2 – Sheet 2). However, the fact that this model does not require this information broadens its applicability. We showed that GRouNdGAN-generated data preserves complex cellular dynamics and patterns of the reference dataset such as lineage trajectories, pseudo-time orderings, and patterns of cell type markers' activation. One consideration in generating realistic and generalizable cells is the number of training examples. In the Bone-Marrow dataset (which had only ~2.7k cells), the performance of all simulators dropped on the test set compared to the training set. When we generated cells using another similar hematopoiesis dataset (Dahlin), but with much larger number of cells (~45k), the testing and training performance difference significantly reduced (Table 3). Similarly, the training and testing performance on all the other datasets that had more than ten thousand cells remained on par with each other, suggesting that a large number of cells improves the realism of the simulated data (which is expected). In spite of this, the cells generated by GRouNdGAN could capture trajectories and pseudo-time orderings not only in the larger Dahlin dataset, but also in the smaller BoneMarrow dataset.

GRouNdGAN imposes user-defined causal regulatory interactions, while excluding those present in the reference dataset, but not in the input GRN. This makes GRouNdGAN-simulated datasets ideal for benchmarking GRN inference algorithms with the input GRN as the ground truth. We benchmarked eight GRN inference algorithms on the simulated BoneMarrow dataset and showed that our results coincide

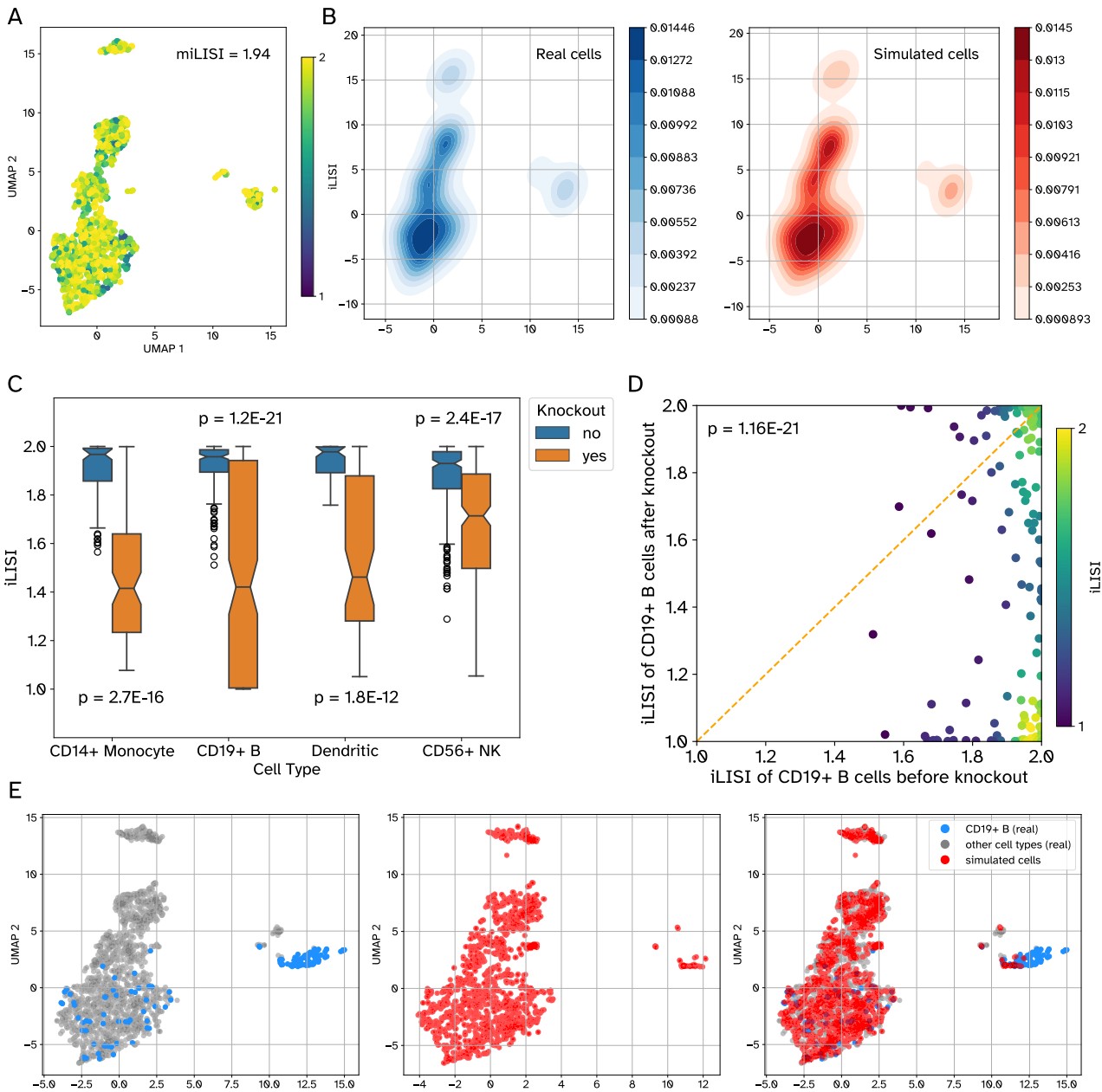

**Fig. 6 | TF knockout experiments and their effect on cell type generation. A** The UMAP shows the distribution of n = 2000 randomly selected cells from the experimental PBMC-All dataset. The color shows the iLISI value of each cell (miLISI = 1.94) calculated from a UMAP embedding, jointly obtained from the experimental cells and the same number of GRouNdGAN-generated cells. TFs were omitted as features when generating UMAP plots. In the simulated data, each gene was regulated by 15 TFs, identified using GRNBoost2. **B** Each plot shows the density of cells in the embedding space: the left plot (blue) corresponds to the real cells and the right plot (red) shows the simulated cells. **C** The boxplots show the distribution of iLISI values of n = 2000 real cells, calculated along with n = 2000 unperturbed simulated cells (blue) and with n = 2000 perturbed simulated cells (orange). For each cell type, top three most differentially expressed TFs were knocked out. The

p-values reported in this figure were calculated using one-sided Wilcoxon signed rank tests. The center horizontal line corresponds to the median. Each box spans the interquartile range, and its limits show the upper (Q3) and lower quartiles (Q1). Whiskers extending from the boxes reach 1.5 times above and below the inter-quartile range. Outliers are shown individually if they fall outside the whiskers. **D** The scatter plot shows the iLISI values of CD19 + B cells calculated along with unperturbed simulated cells (x-axis) and along with perturbed simulated cells (y-axis). The circles correspond to real cells and their colors reflect the density of datapoints in that region. The p-value reported in this figure was calculated using a one-sided Wilcoxon signed rank test. **E** The UMAP embedding of real cells (left), simulated cells after knockout of top TFs of CD19 + B cells (middle), and all together (right). Source data are provided as a Source Data file.

with results reported by BEELINE[19] on curated and experimental benchmarks. This highlights GRouNdGAN's ability to bridge the gap between experimental and simulated datasets for GRN benchmarking without requiring excessive user manipulation. We assessed five of these methods (that did not require pseudo-time) on the PBMC-CTL dataset and observed that their ordering was consistent with the

simulated BoneMarrow data. We then applied them to the PBMC-All dataset, containing multiple distinct cell types. While the general trend was consistent with the other benchmarks, the order of GENIE3 and GRNBoost2 switched. This could be related to the ability of these methods on handling distinct cell types. One should also note that the results of benchmarking on PBMC-All is not conclusive, and follow-up

studies are required to fully characterize the effect of distinct cell types on the performance of GRN inference methods. In particular, GRNs are cell type- and context-specific, and such a study should impose different GRNs on different subpopulations of the dataset. Due to the flexibility provided by GRouNdGAN, one can evaluate a variety of GRN inference methods including supervised methods[49,50] or cell-specific GRN inference methods[48,51]. We included CeSpGRN[48] in our benchmarks, but a comprehensive study that evaluates this and other methods of this class from different perspectives is of great interest.

Since GRouNdGAN generates data based on a structural causal model, it can be used to sample synthetic cells from both observational and interventional distributions. Using this property, we conducted in silico perturbation experiments by knocking out key regulating TFs in various cell types and observed that the targeted cell-states were suppressed in the generated dataset. These results, while preliminary, show potential for developing a method based on GRouNdGAN to predict the results of TF knockout experiments. However, such method would inevitably need access to a high quality GRN underlying the real data, requiring additional data modalities such as (sc)ATAC-seq or TF ChIP-seq.

GRouNdGAN maintains key advantages over existing simulators that incorporate GRNs such as SERGIO and BoolODE. BoolODE, unlike GRouNdGAN, is a reference-free (de novo) and rule-based simulator that relies on user-defined parameters instead of a reference dataset to artificially introduce properties of single-cell data (e.g., dropout, batch effect, existence of distinct populations, and variation among experimental samples and conditions). As a rule-based simulator, it allows for flexible selection of simulation parameters such as choice of noise strength, dropout rate, or kinetic parameters (for mRNA transcription, protein translation and mRNA and protein degradation rates). However, extensive user knowledge is required to leverage this flexibility to avoid generating unrealistic datasets and recent studies have argued that one should use discretion when interpreting results obtained from such de novo simulations[52]. This may explain why some of GRN inference methods that performed best on BoolODE's synthetic benchmarks, were among the worst on the experimental benchmarks of BEELINE. Moreover, as a reference-free simulator, BoolODE is not designed to match a specific reference dataset. As a result, it does not lend itself to benchmarking GRN inference methods in distinct contexts characterized by experimental reference datasets.

On the other hand, SERGIO is a hybrid method that contains elements of reference-free simulators, but also enables matching the simulated cells to a specific reference dataset by adding technical noise. SERGIO simulates data by first generating a "clean" dataset based on SDEs. Then, the user would iteratively fine-tune three noise parameters until the simulated and reference datasets are matched. However, this data-matching procedure puts extra burden on the user and might lead the user to overly rely on fine-tuning technical noise parameters, instead of SDE parameters. Our experiments with SERGIO (described in Supplementary Notes) showed that in spite of our best efforts for distribution matching, there were clear differences between the SERGIO-simulated data and the reference dataset. Another consideration is that over relying on added technical noise to achieve data-matching can potentially corrupt the data, rendering the underlying regulatory patterns unobservable in the final simulated dataset. This might explain the near-random GRN inference performance of various methods reported by SERGIO on their noisy datasets[4].

In contrast to BoolODE and SERIGO, during training, GRouNdGAN's parameters are optimized to minimize any mismatch between real and simulated data distributions automatically. This enables it to replicate scRNA-seq data characteristics (e.g., technical and biological noise) without any user intervention or post-processing. Moreover, GRouNdGAN is trained such that each of its generated genes or TFs mimics the expression patterns of a corresponding gene or TF in the reference dataset, preserving gene/TF identities. In contrast, BoolODE and SERGIO simulate pseudo-genes whose identities cannot be readily mapped to their counterparts in the reference dataset. Gene identity preservation can be attained in these simulators by fine-tuning the SDE parameters until the expression patterns of a given generated gene aligns with a corresponding gene in the reference dataset. This, however, places further burden on the user. Finally, GRouNdGAN learns complex non-linear co-regulatory patterns from the data, instead of relying on simplifying assumptions or user input. In contrast, SERGIO assumes that the combined effect of multiple regulatory TFs is simply the sum of their individual effects. Similarly, BoolODE requires the user to provide a truth table specifying combinatory TF co-regulation rules for each target gene. In contrast, GRouNdGAN's neural network-based generator enables it to implicitly learn complex non-linear co-regulatory patterns that may represent co-expression, co-inhibition, or co-activation of individual genes. This is achieved during the training of GRouNdGAN without user intervention.

Several key points need to be considered when using GRouNdGAN. First, our analyses showed that the architectural and hyper-parameter choices of the presented model result in good performance across different datasets. We performed hyperparameter tuning on one dataset and used the same set of values for all datasets and for different choices of GRNs, obtaining state-of-the-art performance. However, when trained on a completely new dataset, or datasets containing significantly different number of genes and TFs, it is advisable to try different choices of the hyperparameters using a validation set. Second, the architecture of the model includes LSN layers and, as such, the model generates expression values library-size normalized across the considered genes and TFs. Our ablation study (Supplementary Notes) and our experiments showed that the LSN layer improves the GRN imposition, the realism of the generated samples, and stabilizes the training (an observation also made by the scGAN study). In the future iterations of the model, we are planning to remove this restriction to enable generating unnormalized count data. Third, an important consideration for any causal inference problem is that given an observational dataset, its underlying causal graph is not necessarily unique. When benchmarking GRN inference methods, this problem exists regardless of how the ground truth GRN is obtained (even using knockout/knockdown experiments), and a GRN inference algorithm may find a plausible causal GRN, yet not the one that generated the data. Despite this, our analysis using the positive/negative control GRNs showed that GRN inference methods could detect the imposed edges while ignoring others that were unimposed. Based on this observation, we posit that the imposed GRN by GRouNdGAN, while not the only causal GRN capable of generating the simulated dataset, is the most probable one. However, verifying this requires theoretical analyses that are well beyond the scope of this study. Fourth, GRouNdGAN is a generative model and is not a predictive model. As such, its goal is to simulate cells similar to a reference training set (while imposing a user-defined GRN), and not to predict the scRNA-seq profile of cells in a new context or an independent dataset. Given a new dataset, the model needs to be retrained to generate realistic samples.

Finally, although GRouNdGAN can generate data with any GRN, the resemblance of the data to the reference dataset deteriorates if the imposed TF-gene relationships are significantly different from those consistent with the patterns in the reference dataset. The reason is that in such a scenario, generating simulated datapoints resembling the reference training dataset and imposing the GRN act as contradictory requirements. This is a fundamental problem that is not unique to GRouNdGAN and is applicable to any simulator with these two requirements. However, simulators that do not try to impose a causal graph (e.g., scGAN) can focus on generating realistic scRNA-seq samples, even at the expense of disrupting some of the interactions present in the reference dataset (as we observed in Fig. 3 with scGAN). As a result, for GRN inference method benchmarking, we recommend first

finding a set of regulatory edges from the reference dataset (e.g., using GRNBoost2 or even other data modalities) and imposing a selected subset of those in the simulated data. This ensures that the imposed edges are encoded in the simulated data, the unimposed (or undetected) edges are disrupted (as shown in our analyses), and the simulated dataset resembles the reference dataset. It is important to note that the regulatory edges inferred from the reference dataset do not need to be causal (and most likely, some are not); some of them could represent spurious correlations that are observable in the data. In fact, most GRN inference methods do not identify causal edges, yet can be used to form the input GRN. When a subset of the (potentially) non-causal edges of the reference dataset are imposed by GRouNdGAN, they are imposed in a causal manner, representing the *causal* graph underlying the *simulated* data (and not the reference data).

The current version of GRouNdGAN only allows imposing a bipartite GRN and does not support a multi-layer GRN, an assumption that we plan to relax with the future versions of the model. Moreover, we intend to augment the model's architecture in the future to enable imposing a causal graph with three layers of nodes capturing TF-gene-phenotype relationships. This is particularly useful for development and evaluation of phenotype-relevant GRNs, a concept that we have introduced in the past[11]. Additionally, a conditional version of GRouNdGAN can be developed, allowing the user to generate cells of specific cell type or on a specific stage along a lineage trajectory. Finally, by incorporating DAG structural learning[53], we intend to enable GRouNdGAN to infer a GRN while simultaneously learning to mimic the reference dataset.

## Methods

### Datasets and preprocessing

We downloaded the human peripheral blood mononuclear cell (PBMC Donor A) dataset containing the single-cell gene expression profiles of 68579 PBMCs represented by UMI counts from the 10x Genomics (https://support.10xgenomics.com/single-cell-gene-expression/datasets/1.1.0/fresh_68k_pbmc_donor_a). This dataset contains a large number of well-annotated cells and has been used by other models to generate synthetic scRNA-seq samples[24]. We also downloaded the scRNA-seq profile of 2730 cells corresponding to differentiation of hematopoietic stem cells to different lineages from mouse bone marrow[34] ("Bone-Marrow" dataset) from Gene Expression Omnibus (GEO) (accession number: GSE72857). We also obtained another haematopoietic dataset corresponding to the scRNA-seq (10x Genomics) profiles of 44,802 mouse bone marrow hematopoietic stem and progenitor cells (HSPCs) differentiating towards different lineages from GEO (accession number: GSE107727)[46] (called Dahlin dataset here). Finally, we obtained the batch-corrected scRNA-seq (10x Genomics) profile of 136,147 cells corresponding to malignant cells as well as cells in the tumor microenvironment (called Tumor-All dataset here) from 20 fresh core needle biopsies of follicular lymphoma patients from (https://cellxgene.cziscience.com/collections/968834a0-1895-40df-8720-666029b3bbac)[47].

We followed pre-processing steps similar to those of scGAN and cscGAN[24] using scanpy version 1.8.2[45]. In each dataset, cells with non-zero counts in less than ten genes were removed. Similarly, genes that only had nonzero counts in less than three cells were discarded. Top 1000 highly variable genes were selected using the dispersion-based method described by Satija et al.[54]. Finally, library-size normalization was performed on the counts per cell with a library size equal to 20,000 in order to be consistent with previous studies[24]. See Supplementary Data 1 - Sheet 3 for the number of cells and highly variable genes and TFs present in each final dataset.

### GRouNdGAN's model architecture

GRouNdGAN's architecture consists of 5 components, each implemented using separately parameterized neural networks: a causal controller, target generators, a critic, a labeler, and an anti-labeler (Fig. 1C).

**Causal controller.** The role of causal controller is to generate the expression of TFs that causally control the expression of their target genes based on a user-defined gene regulatory network (Fig. 1C). To achieve this, it is first pre-trained (Fig. 1B) as the generator of a Wasserstein GAN with gradient penalty (WGAN-GP) (see Supplementary Notes for the formulation of the Wasserstein distance). GANs are a class of deep learning models that can learn to simulate non-parametric distributions[55]. They typically involve simultaneously training a generative model (called the "generator") that produces new samples from noise, and its adversary, a discriminative model that tries to distinguish between real and generated samples (called the "discriminator"). The generator's goal is to generate samples so realistic that the discriminator cannot determine whether it is real or simulated (an accuracy value close to 0.5). Through adversarial training, the generator and discriminator receive feedback, allowing them to co-evolve in a symbiotic manner.

The main difference between a WGAN and a traditional GAN is that in the former, a Wasserstein distance is used to quantify the similarity between the probability distribution of the real data and the generator's produced data (instead of Kullback–Leibler or Jensen–Shannon divergences). Wasserstein distance has been shown to stabilize the training of WGAN without causing mode collapse[32]. The detailed formulation of the Wasserstein distance used as the loss function in this study is provided in Supplementary Notes. In addition, instead of a discriminator, WGAN uses a "critic" that estimates the Wasserstein distance between real and generated data distributions. In our model, we added a gradient penalty term for the critic (proposed by Gulrajani et al.[56] as an alternative to weight clipping used in the original WGAN) in order to overcome vanishing/exploding gradients and capacity underuse issues.

In the pretraining step, we trained a WGAN-GP with a generator (containing an input layer, three hidden layers, and an output layer), a library-size normalization (LSN) layer[24], and a critic (containing an input layer, three hidden layers, and an output node). A noise vector of length 128, with independent and identically distributed elements following a standard Gaussian distribution, was used as the input to the generator. The output of the generator was then fed into the LSN layer to generate the gene and TF expression values. The details of hyper-parameters and architectural choices of this WGAN-GP are provided in Supplementary Table 1. Although we were only interested in generating expression of TFs using the generator of this WGAN-GP (in the second step of pipeline), the model was trained using all genes and TFs to properly enforce the library-size normalization. Once trained, we discarded the critic and the LSN layer, froze the weights of the generator and used it as the "causal controller"[31] to generate expression of TFs (Fig. 1C).

**Target generators.** The role of target generators is to generate the expression of genes causally regulated by TFs based on the topology of a GRN. Consider a target gene $Gj$ regulated by a set of $TFs$ : $\{TF_1, TF_2, \ldots, TF_n\}$. Under the causal sufficiency assumption and as a result of the manner by which TFs' expressions are generated from independent noise variables, we can write $E_{Gj} = f_{Gj}(E_{TF_1}, E_{TF_2}, \ldots, E_{TFn}, N_{Gj})$ and $E_{TFi} = f_{TFi}(N_{TFi})$ for $i = 1, 2, \ldots, n$, where $E$ represents expression, $N$ represents a noise variable, and $f$ represents a function (to be approximated using neural networks). All noise variables are jointly independent. Following the theoretical and empirical results of CausalGAN[31], we can use feedforward neural networks to represent functions $f$ by making the generator inherit its neural network connections from the causal GRN. To achieve this, we generate each gene in the GRN by a separate generator such that target gene generators do not share any neural connections (Fig. 1C).

As input, the generator of each gene accepts a vector formed by concatenating a noise variable and a vector of non-library size normalized TF expressions from the causal controller, corresponding to the TFs that regulate the gene in the imposed GRN (i.e., its parents in the graph). The expression values of TFs and the generated values of target genes are arranged into a vector, which is passed to an LSN layer for normalization. We used target generators with three hidden layers of equal width. The width of the hidden layers of a generator is dynamically set as twice the number of its regulators (the noise variable and the set of regulating TFs). If the imposed GRN is relatively dense and contains more than 5000 edges, we set the depth to 2 and the width multiplier to 1 to be able to train on a single GPU. The details of hyperparameters and architectural choices are provided in Supplementary Table 2.

In practice, generating each target gene's expression using separate neural networks introduces excess overhead. This is because in every forward pass, all target genes' expressions must be first generated before collectively being sent into the LSN layer. As a result, instead of parallelizing the generation of each target gene's expression (which due to the bottleneck above does not provide a significant computational benefit), we implemented target generators using a single large sparse network. This allows us to reduce the overhead and the training time to train the model on a single GPU, and to benefit from GPU's large matrix multiplication. We mask weights and gradients to follow the causal graph, while keeping the generation of genes independent from each other. From a logical standpoint, our implementation has the same architecture described earlier, but is significantly more computationally efficient. See Supplementary Notes for details.

**Critic.** Similar to a traditional WGAN, the objective of GRouNdGAN's critic (Fig. 1C) is to estimate the Wasserstein distance between real and generated data distributions. We used the same critic architecture as the WGAN-GP trained in the first stage.

**Labeler and anti-labeler.** Although the main role of the target generators is to produce realistic cells to confuse the critic, it is crucial that they rely on the TFs' expression (in addition to noise) in doing so. One potential risk is that the target generators disregard the expression of TFs and solely rely on the noise variables. This is particularly probable when the imposed GRN does not conform to the underlying gene expression programs of real cells in the reference dataset; in such a scenario, and to make realistically looking simulated cells, it is more convenient for a WGAN to simply ignore the strict constraints of the GRN and solely rely on noise.

To overcome this issue, we used auxiliary tasks and neural networks known as "labeler" and "anti-labeler"[31]. The task of these two networks is to estimate the causal controller's TF expressions (here called labels) from the target genes' expressions alone, by minimizing the squared L2 norm between each element's TF estimates and their true value. More specifically, the corresponding loss for a batch of size $N_{batch}$ is of the form $\frac{1}{N_{batch}} \sum_{i=1}^{N_{batch}} ||\hat{y}_i - y_i||_2^2$, where $\hat{y}_i$ is the estimated vector of TF expression values generated by the labeler or anti-labeler. For anti-labeler, $y_i$ corresponds to the TF expression values outputted by the causal controller; for labeler, this vector can also correspond to TF expression values from the real training data. This resembles the idea behind an autoencoder and ensures that the model will not disregard the expression of TFs in generating the expression of their target genes. The anti-labeler is trained solely based on the outputs of the target generators, while the labeler utilizes both the outputs of target generators and the expression of real cells (Fig. 1C). They are both implemented as fully connected networks with a width of 2000 and a depth of 3 and optimized using the AMSGrad[57] algorithm. Each layer, except the last one, utilizes a ReLU activation function and batch normalization. In addition to WGAN-GP losses, we add labeler and anti-

labeler losses to the generator to minimize both. This is different from the approach used in CausalGAN, where the anti-labeler's loss is maximized in the early training stages to ensure that the generator doesn't fall into label-conditioned mode collapse. In GRouNdGAN, the causal controller is pretrained and generates continuous labels (TFs expression) and does not face a similar issue. As a result, we instead minimized the loss of the anti-labeler from the beginning and as such the labeler and anti-labeler both act as auxiliary tasks to ensure the generated gene expression values take advantage of TFs expression.

### Training procedure and hyperparameter tuning

We follow a two-step training procedure comprising of training two separate WGAN-GPs (Fig. 1) to train GRouNdGAN. The generators and critics in both GANs are implemented as fully connected neural networks with rectified linear unit (ReLU) activation functions in each layer, except for the last layer of the critic. The weights were initialized using He initialization[58] for layers containing ReLU activation and Xavier initialization[59] for other layers (containing linear activations). We used Batch Normalization[60] to normalize layer inputs for each training minibatch, except for the critic. This is since using it in the critic invalidates the gradient penalty's objective, as it penalizes the norm of the critic's gradient with respect to the entire batch rather than to inputs independently. An LSN layer[24] was used in both WGAN-GPs (Fig. 1B, C) to scale counts in each simulated cell to make it consistent with the library size of input reference dataset. This normalization results in a dramatic decrease in convergence time and smooths training by mitigating the inherent heterogeneity of scRNA-seq data.

We would like to point out that both steps of training (i.e., the pretraining of Fig. 1B and training of Fig. 1C) are performed on the exact same training set, and when data corresponding to a new dataset is to be generated, these steps need to be repeated. The goal of the first step is to train the causal controller to learn the distribution of training set and generate realistic TF expression values (without imposing TF-gene relationships). The goal of the second step is to use the TF expression values generated by the trained causal controller to generate expression of target genes while imposing the TF-gene causal relationships.

GRouNdGAN solves a min-max game between the generator ($f_g$) and the critic ($f_c$), with the following objective function:

$$\min_{f_g} \max_{||f_c||_L \leq 1} \mathbb{E}_{x \sim \mathbb{P}_r} f_c(x) - \mathbb{E}_{x \sim f_g(\mathbb{P}_{noise})} f_c(x) \tag{1}$$

The training objective of the critic involves maximizing the difference between the average score assigned to real ($\mathbb{E}_{x \sim \mathbb{P}_r} f_c(x)$) and generated samples ($\mathbb{E}_{x \sim f_g(\mathbb{P}_{noise})} f_c(x)$) with respect to its parameters following Eq. (1). On the contrary, the generator attempts to minimize the average score that the critic assigns to real and generated samples. Through this adversarial game, both the critic and generator co-evolve and the generator learns a mapping $f_g$ from a simple standard Gaussian noise distribution ($\mathbb{P}_{noise}$) to a distribution $f_g(\mathbb{P}_{noise})$ that approximates the real data distribution $\mathbb{P}_r$. An important point to consider is that the critic does not directly compute the Wasserstein distance (mathematically defined in Supplementary Notes - Details regarding the Wasserstein distance). Instead, it is encouraged to provide a meaningful estimate of the Wasserstein distance between the distribution of real data ($\mathbb{P}_r$) and the distribution of generated data ($f_g(\mathbb{P}_{noise})$) through its training process.

We alternated between minimizing the generator loss for one iteration and maximizing the critic loss for five iterations. We employed the AMSGrad[57] optimizer with the weight decay parameters $\beta_1 = 0.5$, $\beta_2 = 0.9$ and employed an exponentially decaying learning rate for the optimizer of both the critic and generator.

The hyperparameters were tuned using a validation set consisting of 1000 cells from the PBMC-CTL dataset (Supplementary Tables 1 and 2) based on the Euclidean distance and the RF AUROC

score, which were consistently in accord. The same hyperparameters were used for all other analyses and datasets.

## Causal GRN preparation

This section describes the creation of the causal graph inputted to GRouNdGAN used to impose a causal structure on the model. GRouNdGAN accepts a GRN in the form of a bipartite directed acyclic graph (DAG) as input, representing the relationship between TFs and their target genes. In this study, we created the causal graph using the 1000 most highly variable genes and TFs identified in the preprocessing step (Supplementary Data 1 - Sheet3). First, the set of TFs among the highly variable genes were identified based on the AnimalTFDB3.0 database[61] and a GRN was inferred using GRNBoost2[8] (with the list of TFs provided) from the training reference dataset. It is important to note that the regulatory edges identified from the reference dataset using GRNBoost2 are not necessarily causal edges (and they do not need to be for the purpose of forming the input GRN), but they are consistent with the patterns of the data. However, when this (potentially non-causal) GRN is imposed by GRouNdGAN, it is imposed in a causal manner and represents the causal data generating graph of the simulated data (and not the reference data).

## Evaluation of the resemblance of real and simulated cells

We evaluated all models using held-out test sets containing randomly selected cells from each reference dataset (500 cells from Bone-Marrow and 1000 cells for all other datasets) (see Supplementary Data 1 - Sheet 3 for other statistics about the datasets). To quantify the similarity between real and generated cells, we employed various metrics. For each cell represented as a datapoint in a low dimensional embedding (e.g., t-SNE or UMAP), the integration local inverse Simpson's Index (iLISI)[36] captures the effective number of datatypes (real or simulated) to which datapoints of its local neighborhood belong based on weights from a Gaussian kernel-based distributions of neighborhoods. The miLISI is the mean of all these scores and in our study ranges between 1 (poor mixing of real and simulated cells) and 2 (perfect mixing of real and simulated cells). Additionally, we calculated the cosine and Euclidean distances of the centroids of real cells and simulated cells, where the centroid was obtained by calculating the mean along the gene axis (across all simulated or real cells).

To estimate the proximity of high-dimensional distributions of real and simulated cells without creating centroids, we used the maximum mean discrepancy (MMD)[35]. Given two probability distributions $p$ and $q$ and a set of independently and identically distributed (i.i.d.) samples from them, denoted by $X$ and $Y$, MMD with respect to a function class $\mathcal{F}$ is defined as

$$\mathrm{MMD}[\mathcal{F}, X, Y] := \sup_{f \in \mathcal{F}} \left( \mathbb{E}_x[f(x)] - \mathbb{E}_y[f(y)] \right), \quad (2)$$

where sup refers to supremum and $\mathbb{E}$ denotes expectation. When the MMD function class $\mathcal{F}$ is a unit ball in a reproducing kernel Hilbert space (RKHS) $\mathcal{H}$ with kernel $k$, the population MMD takes a zero value if and only if $p = q$ and a positive unique value if $p \neq q$. The squared MMD can be written as the distance of mean embeddings $\mu_p$, $\mu_q$ of distributions $p$ and $q$, which can be expressed in terms of kernel functions:

$$MMD^2[F, p, q] = ||\mu_p - \mu_q||_H^2 \quad (3)$$

$$= \mathbb{E}_{x,x'}\left[k(x,x')\right] - 2\mathbb{E}_{x,y}\left[k(x,y)\right] + \mathbb{E}_{y,y'}\left[k(y,y')\right]$$

Following existing implementations of MMD in the single-cell domain[24,62], we chose a kernel that is the sum of three Gaussian kernels

to increase sensitivity of the kernel to a wider range:

$$k(x,y) = \sum_i \left( \frac{||x-y||^2}{\sigma_i^2} \right), i = 1,2,3 \quad (4)$$

where $\sigma_i$ denote standard deviations and were chosen to be the median of the average distance between a point to its 25 nearest neighbors divided by factors of 0.5, 1, and 2 in the three kernels, respectively.

We also used a random forests (RF) classifier and used its area under the receiver operating characteristic (AUROC) curve to determine whether the real and simulated cells can be distinguished from each other. Consistent with previous studies[24,63], we first performed a dimensionality reduction using principal component analysis (PCA) and used the top 50 PCs of each cell as the input features to the RF model, which improves the computational efficiency of this analysis. The RF model was composed of 1000 trees and the Gini impurity was used to measure the quality of a split.

## Baseline simulator models

We compared the performance of GRouNdGAN to scDESIGN2[25], SPARSim[26], and three GAN-based methods: scGAN[24], cscGAN with projection-based conditioning[24], and a conditional WGAN (cWGAN). The cWGAN method conditions by concatenation following the cGAN framework[64]. More specifically, it concatenates a one-hot encoded vector (representing the cluster number or cell type) to the noise vector input to the generator and cells forwarded to the discriminator. We did not train the cWGAN or cscGAN on the PBMC-CTL dataset, since it contains only one cell type. For the PBMC-All and the Bone-Marrow dataset, we trained all models above. Additionally, we simulated data using scDESIGN2 and SPARSim with and without cell cluster information, as they allow providing such side information in their training.

To train models that utilized cell cluster information, we performed Louvain clustering and provided the cluster information and ratio of cells per cluster during training. Clustering was done by following the cell ranger pipeline[33], based on the raw unprocessed dataset (and independent of the pre-processing steps described earlier for training simulators). First, genes with no UMI count in any of the cells were removed. Then the gene expression profile of each cell was normalized by the total UMI of all (remaining) genes, and highly variable genes were identified. The gene expression profile of each cell was then re-normalized by the total UMI of retained highly variable genes, and each gene vector (representing its expression across different cells) was z-score normalized. Given the normalized gene expression matrix above, we found top 50 principal components (PCs) using PCA analysis. These PCs were then used to compute a neighborhood graph with a local neighborhood size of 15, which was used in Louvain clustering. We ran the Louvain algorithm with a resolution of 0.15.

For SPARSim, we set all sample library sizes to 20000 and estimated gene expression level intensities and simulation parameters by providing it with both raw and normalized count matrices. When cell cluster information was provided, distinct SPARSim simulation parameters were estimated per cell for each cluster. scDESIGN2 accepts input matrices where entries are integer count values; we thus performed rounding on the expression matrix before fitting scDESIGN2. With cluster information provided, a scDESIGN2 model was fit separately for cells of each cluster, and similar to conditional GANs, the ratio of cells per cluster was provided to the method.

## In silico perturbation experiments using GRoundGAN

To perform perturbation experiments using GRoundGAN, we put the trained model in a deterministic mode of operation. This is necessary to ensure that the perturbation experiments are performed on the

same batch (i.e., replicate) of generated cells to form matched case/control experiments. To do this, we performed a forward pass through the generator and then saved the input noise to the causal controller, the input noise to the target generators, the TF expression values generated by the causal controller, and the per-cell scaling factor of the LSN layer. Subsequent passes through the generators used the saved parameters so that ensuing runs always output the same batch of cells (instead of generating new unmatched cells).

### Trajectory inference and pseudo-time analysis
Following official PAGA tutorial for the BoneMarrow dataset (https://github.com/scverse/scanpy-tutorials/blob/master/paga-paul15.ipynb), we used (Partition-based graph abstraction) PAGA[38] for trajectory inference and analysis. We built force-directed graphs[65] (with ForceAtlas2[66]) using the top 20 principal components of the data (using principal component analysis or PCA) and a neighborhood graph of observations computed using UMAP (to estimate connectivities). We next denoised the graph by representing it in the diffusion map space and computed distances and neighbors as before using this new representation. After denoising, we then ran the Louvain clustering algorithm with a resolution of 0.6. Finally, we ran the PAGA algorithm on the identified clusters and used the obtained graph to initialize and rebuild the force-directed graph.

### GRN inference methods
In our GRN benchmarking analysis, we focused on eight GRN inference algorithms: GENIE3[9], GRNBoost2[8], PPCOR[37], PIDC[10], LEAP[12], SCODE[13] and SINCERITIES[14], which were used in the BEELINE study[19], as well as CeSpGRN[48]. Of these methods, LEAP requires pseudo-time ordering of cells, while SCODE and SINCERITIES require both pseudo-time ordering and pseudo-time values. Since not all algorithms inferred the edge directionality or its sign (activatory or inhibitory nature), we did not consider these factors in our analysis to be consistent among different models.

For the methods available in the BEELINE study, we ran them as docker containers using the docker images provided by BEELINE's GitHub (https://github.com/Murali-group/Beeline)[19] with the default parameters used in BEELINE. These methods were applied to nine datasets simulated by GRouNdGAN and scGAN, and the original training real dataset corresponding to PBMC-CTL, PBMC-All, and BoneMarrow. To ensure consistency, the same number of cells as the real training set were simulated using GRouNdGAN and scGAN for each dataset: $n = 19773$ for PBMC-CTL, $n = 67579$ for PBMC-All, and $n = 2230$ for BoneMarrow. Number of genes and TFs present in the GRN for each dataset is provided in Supplementary Data 1 - Sheet 3. To benchmark algorithms requiring pseudo-time ordering of cells, we computed the pseudo-times of GRouNdGAN-simulated data (based on the BoneMarrow dataset) using PAGA[38] and diffusion pseudotime[39], following the methodology described earlier. In the GRN inference benchmark analysis, we did not provide the list of TFs to GRNBoost2 to make it consistent with other GRN inference methods.

We also included CeSpGRN, which is a cell-specific GRN inference method. Since this method first generates one GRN for each cell, it requires a high amount of memory to run. As a result, we were only able to benchmark it using a subset of data consisting of only $n = 1000$ cells (for any of the three datasets) and 100 genes (we did not change the number of TFs, or the GRN edges connecting them to the considered genes). Following the method described in the original study, we then averaged the total absolute edge weights across all cells to form a consensus GRN using CeSpGRN.

### Statistics and reproducibility
The sample sizes were selected by original studies producing datasets used for training GRouNdGAN. Preprocessing steps are described in the datasets and preprocessing section. Other than standard cell-level and gene-level filtering, there were no data exclusions. Statistical tests used for each analysis are described in the corresponding sections and include Wilcoxon signed rank and Mann Whitney U tests.

### Reporting summary
Further information on research design is available in the Nature Portfolio Reporting Summary linked to this article.

## Data availability
The PMBC reference dataset is available from the 10x Genomics repository (corresponding to healthy donor A) from the following link: https://support.10xgenomics.com/single-cell-gene-expression/datasets/1.1.0/fresh_68k_pbmc_donor_a? (related to PBMC-All, PBMC-CTL, and PBMC-NaiveT). The BoneMarrow reference dataset is available in the Gene Expression Omnibus (GEO) repository under accession number GSE72857. The Dahlin reference dataset is available in GEO (accession number: GSE107727). The Tumor-All (and Tumor-malignant) dataset is available from cellxgene (https://cellxgene.cziscience.com/collections/968834a0-1895-40df-8720-666029b3bbac). Simulated data corresponding to PBMC-All, PBMC-CTL, BoneMarrow, Dahlin, Tumor-All, and Tumor-malignant and their corresponding imposed GRNs are provided on GRouNdGAN's website (https://emad-combine-lab.github.io/GRouNdGAN/benchmarking) and can be used for benchmarking different GRN Inference methods. Source data are provided with this paper.

## Code availability
Our implementation and evaluation of GRouNdGAN in Python 3.9.6 using the PyTorch framework[67] along with a tutorial is freely available under the GNU Affero General Public License v3.0 on GitHub (https://github.com/Emad-COMBINE-lab/GRouNdGAN) which is archived in Zenodo under record number 11068246[68] (https://doi.org/10.5281/zenodo.11068246). The repository also contains a Docker image for reproducibility and our PyTorch implementation of scGAN, projection-based conditioning cscGAN[24] and a variation of cscGAN that uses conditioning by concatenation (cWGAN). More information about the installation, a detailed tutorial, and simulated datasets that can be readily used are provided in (https://emad-combine-lab.github.io/GRouNdGAN).

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

## Acknowledgements

This work was supported by grants from Natural Sciences and Engineering Research Council of Canada (NSERC) [RGPIN-2019-04460] (A.E.), Government of Canada's New Frontiers in Research Fund (NFRF) [NFRFE-2019-01290] (A.E.), Canada Foundation for Innovation (CFI) JELF [project 40781], and McGill Initiative in Computational Medicine (A.E.). This research was enabled in part by support provided by the Digital Research Alliance of Canada (https://alliancecan.ca) and Calcul Québec (www.calculquebec.ca). *This publication is part of the Human Cell Atlas –*www.humancellatlas.org/publications.

## Author contributions

All authors contributed to the design of the study and the algorithm, the analyses, and the writing of the manuscript. A.E. supervised the study. Y.Z. implemented the model and performed the statistical analyses. Y.Z., A.T., and A.E. read and approved the final manuscript.

## Competing interests

The authors declare no competing interests.
