## [Peer Review File · Nature Communications]

GRouNdGAN: GRN-guided simulation of single-cell RNA-seq data using causal generative adversarial networksReviewer #1 (Remarks to the Author):

This paper proposes a deep-learning based generative model GRouNdGAN to simulate single-cell RNA-sequencing (scRNA-seq) expression data based on given gene regulatory networks (GRNs). The architecture of GRouNdGAN is based on CausalGAN (Causal Generative Adversarial Network), which is designed as a novel generator to simulate scRNA-seq gene expression while preserving causal TF-target interactions. The authors benchmarked GRouNdGAN and other simulators based on two publicly available scRNA-seq datasets (PMBCs and BoneMarrow) and showed GRouNdGAN was more robust in simulating both steady-state and transient-state single-cell datasets. It can preserve gene identities, cell trajectories, pseudo-time ordering, and technical and biological noise. This work is interesting and timely, I believe it could be much useful in developing GRN algorithms. I still have several questions need to be addressed.

Major comments

1. The target generator (used to generate target gene expression) consists of n (the number of the target genes) independent fully connected networks, which are time consuming due to a large number of n . The manuscript claims to replace them by a single large sparse network in the target generator to reduce the training time. However, I cannot find its architecture and how it works in model training. In the source code, I found `masked_causal_generator.py` (line 155, the function of `_create_generator()`) to implement this sparse network using three mask matrices, which is not described in the paper.
2. The supplementary table notations lack clarity. The authors provided three Excel files corresponding to Supplementary Table S4-S6. However, within each file, there are multiple tabs, and the associations are not well-defined. The Tables S7 and S8, mentioned in the paper, are also elusive and cannot be located."
3. In 'Performance w.r.t. Resemblance' table, the paper asserts that GRouNdGan's performance on BoneMarrow is comparable with other datasets, yet I observed a RF AUROC of 0.75 on the test data, indicating subpar performance. Also, could you please provide insight into the substantial performance difference of GRouNdGan between its training and testing data for the BoneMarrow dataset?
4. In data simulation on PBMC-ALL, is it understood correctly that identical GRNs were applied across distinct cell types? I'm uncertain whether this might influence the expression of cell-type marker genes. Could you elucidate how your experimental setup ensures the consistency of cell type annotations?
5. Line 218, did the same experiments also conduct on PBMC-ALL? I'm curious whether the diversity of cell types might impact the reconstruction of GRNs.
6. I suggest to involve more tools for cell-type-specific and cell-specific gene regulatory network construction, such as CNNC, DeepDRIM, CeSpGRN, LocaTE et al.
7. It is better to show the performance of GRouNdGan if the number of TFs exceeds 15.

Minor comments

1. The authors indicated the generation of five replicates, each containing 1000 cells, to assess the robustness of the simulated data. Could you provide specific details about how these replicates were utilized to create Figure 2? For instance, were average values employed?
2. It might enhance clarity to include the exact formulations of the cost functions for both the Labeler and anti-labeler, rather than a general mention of "minimizing the squared L2 norm."
3. In Figure 1, it appears that the input for the Critic module should be a vector rather than an entire matrix.
4. In Line 150, when referring to "control," does this term imply the actual cells within the training/test sets? Furthermore, could you elaborate on the interpretation of the values in the row labeled "control" in Table 1?

Code and Software Review

The author releases the source code of GRouNdGAN on Github and provides the guide to install and use it (<https://emad-combine-lab.github.io/GRouNdGAN/index.html>) with the demo dataset. I believe the installation using Docker and Singularity approaches should be okay (not tested), but it is not easy to run the program on HPC without root access.

I had some issues when installing GRouNdGAN:

1. Although the authors suggest users to utilize `Pyenv` to manage Python versions, it is only

designed for managing multiple Python versions on the same node. It should be coupled with the plugin `Pyenv-virtualenv` (<https://github.com/pyenv/pyenv-virtualenv>) to allow users to keep code environments separately. It makes my packages from Conda crashed if I install GRounNdGAN following the instructions.

2. In addition, `source venv/bin/activate` can activate the created environment, but the installation path is still my user Python path.

3. When I ran `pip install -r requirements.txt`, the packages were installed into my python user path rather than the required path.

I have tried to fix the problems by myself using the steps below:

1. Install Pyenv (<https://github.com/pyenv/pyenv#basic-github-checkout>) and set the environment variable by hand; Install `Pyenv-virtualenv` (<https://github.com/pyenv/pyenv-virtualenv>)

2. Run `pyenv install 3.9.6` to get a specific python version; Run `pyenv virtualenv 3.9.6 venv` to create the environment; Run `pyenv activate venv` to activate the environment; Run `echo \$PATH`, `which pip` and `which python` to check the path, pip tool, and python interpreter, respectively.

3. Run `pip install -r requirements.txt`

4. ModuleNotFoundError: No module named `packaging` occurred when installing `docopt==0.6.2`.

Manually install it to solve the error: Run `pip install packaging --target ~/.pyenv/versions/3.9.6/envs/venv/lib/python3.9/site-packages`

5. AttributeError: module 'numpy' has no attribute 'version' occurred when installing `statsmodels==0.11.0rc2`;

Manually install it to solve the error: Run `pip install numpy --target ~/.pyenv/versions/3.9.6/envs/venv/lib/python3.9/site-packages`

All required packages can be downloaded successfully, but I encountered other problems:

1. ERROR: Could not build wheels for docopt, pynndescent, session-info, sinfo, statsmodels, termcolor, tfrecord, umap-learn, which is required to install pyproject.toml-based projects
Manually install the setuptools and wheel packages: Run `pip install setuptools wheel --target ~/.pyenv/versions/3.9.6/envs/venv/lib/python3.9/site-packages --upgrade`

2. ModuleNotFoundError: No module named `_ctypes`.

To solve this error, the `libffi-dev` is required. And, it needs root access, going back to the beginning. I did not successfully install GRounNdGAN eventually.

Reviewer #2 (Remarks to the Author):

I co-reviewed this manuscript with one of the reviewers who provided the listed reports as part of the Nature Communications initiative to facilitate training in peer review and appropriate recognition for co-reviewers.

Reviewer #3 (Remarks to the Author):

Gene regulatory network (GRN) inference is crucial in many aspects of biological research, and the key challenge for this task is to obtain gold standard ground truth GRNs as well as the

corresponding realistic dataset. In the present manuscript, the authors aim to generate scRNA-seq data based on known GRN. This is a computational biology problem of importance, and a successful framework will have widely application scenario. However, current version of GRouNdGAN is not proved to be as powerful as described in the Abstract. More experiments and descriptions are required.

General comments:

1. Details of GRouNdGAN architecture and training procedure.

a). Please provide more details for pre-training in Fig. 1A. In the Critic module, how to calculate the Wasserstein Distance between the real scRNA-seq data and the generated expression vectors of TFs and genes? In addition, in my opinion, the Causal Controller learned the distribution of scRNA-seq data but not regulatory relationships between TFs and genes (which I will discuss below). Thus, if the pre-training scRNA-seq dataset contain different cell types, that is containing different data distribution, it is possible that the pre-train model will not converge.

b). As described in Methods, the imposed GRN is a form of bipartite directed acyclic graph (DAG). In this DAG, the number of genes is fixed to 1000 based on the most highly variable genes, and TFs are identified from these variable genes based on the AnimalTFDB3.0 database. GRouNdGAN trained Target generators for each TF. However, highly variable genes are different across different datasets, resulting in the dynamic TFs in DAGs. How to deal with the dynamic TFs while GRouNdGAN contains fixed Target generators or will GRouNdGAN train different Target generators for new dataset?

c). As described in Discussion, the authors pointed out that "The current version of GRouNdGAN only allows imposing a bipartite GRN and does not support regulation of TFs by other TFs". So, in DAGs, if the 1000 highly variable genes contain TF genes, will TF genes be discarded?

d). In line 168, "generate samples from different cell types" means generate scRNA-seq data containing different cell types or generate scRNA-seq data of a different cell type which is not appeared in training set? If cell type information is available a priori, how to import this priori knowledge into GRouNdGAN?

e). How exactly did GRouNdGAN solve the challenge that when cell type information is unavailable? The authors only described the results on two datasets (human PBMC and mouse BoneMarrow) but did not provide the details how they performed.

f). Figure 1 is rough. Neural networks diagrams are redundant while details for framework input/output and training process are inadequate. I suggest the authors discuss with other colleagues and redraw it.

2. Whether the framework learns regulatory relationships is debatable.

a). As discussed in Comment 1.a, the three-layer neural networks learn the gene expression distribution rather than regulatory relationships. There are inherent shortcomings to the assumption that gene expression patterns correspond to regulatory relationships (Aditya Pratapa et al., Nat. Meth., 2020; Mark W.E.J Fiers et al., Brief Funct Genomics, 2018; Hao Li et al., Nat. Mach. Intell., 2022). To investigate the causal interactions between TFs and genes, multiple modalities data such as (sc)ATAC-seq, TF ChIP-seq are required. Thus, module description of Causal Controller in GRouNdGAN is not proper. This limitation for inferring GRN using scRNA-seq data should be discussed.

b). The authors showed the ability of GRouNdGAN in imposing the input causal GRN by performing in-silico TF knockout experiments. Further, the authors showed that imposed GRN can be reconstructed from the simulated data using existing methods. Above tests are not sufficient to demonstrate that GRN information is integrated the generated scRNA-seq data. As discussed in Comment 2.a, neural networks can learn the expression pattern from scRNA-seq data and are possible to be powerful to generate simulated data without any GRN information. Model training with completely random GRNs and with no GRN imposed are necessary to test this point.

3. Performance evaluation is insufficient.

a). Model stability. It is necessary to test the consistency of multiple generated scRNA-seq for the same cell type.

b). The authors used three scRNA-seq datasets to train the model, and also used these datasets to test it. It is necessary to test the performance of GRouNdGAN on new datasets with GRNs information, which are not appeared in current training datasets. Specially, gene expression patterns as well as GRNs of tumor cells are often different from normal cells, the performance evaluation of GRouNdGAN on tumor data is needed.

c). Pseudo-time analysis and In-silico perturbation experiments are also needed to be conducted on another new dataset.

Minors:

1. Line 46, full name of DREAM.
2. Line 123, Typo "scRAN-seq".
3. Line 140, "red and blue" inconsistency between figures and texts.

Reviewer #1 (Remarks to the Author):

Comment: “This paper proposes a deep-learning based generative model GRouNdGAN to simulate single-cell RNA-sequencing (scRNA-seq) expression data based on given gene regulatory networks (GRNs). The architecture of GRouNdGAN is based on CausalGAN (Causal Generative Adversarial Network), which is designed as a novel generator to simulate scRNA-seq gene expression while preserving causal TF-target interactions. The authors benchmarked GRouNdGAN and other simulators based on two publicly available scRNA-seq datasets (PMBCs and BoneMarrow) and showed GRouNdGAN was more robust in simulating both steady-state and transient-state single-cell datasets. It can preserve gene identities, cell trajectories, pseudo-time ordering, and technical and biological noise. This work is interesting and timely, I believe it could be much useful in developing GRN algorithms. I still have several questions need to be addressed.”

Response: We thank the reviewer for their insightful comments and suggestions. In what follows, we have provided detailed responses and new analyses, which we believe will address your questions. Among other analyses, we are now reporting results on data from two additional independent studies to further augment the analyses.

Major comments:

Comment: “1. The target generator (used to generate target gene expression) consists of n (the number of the target genes) independent fully connected networks, which are time consuming due to a large number of n . The manuscript claims to replace them by a single large sparse network in the target generator to reduce the training time. However, I cannot find its architecture and how it works in model training. In the source code, I found `masked_causal_generator.py` (line 155, the function of `_create_generator()`) to implement this sparse network using three mask matrices, which is not described in the paper.”

Response: The reviewer correctly identified `masked_causal_generator.py` to contain the implementation of the `CausalGenerator` class which implements the target generators. `CausalGenerator` refers to the generator of GRouNdGAN containing the causal controller and target generators which this class encapsulates. `_create_generator` is a method called within the `CausalGenerator` constructor that builds the target generators.

As mentioned in Methods, implementing each target generator as a separate neural network significantly increases the computational cost and introduces excess overhead due to the LSN layer. Hence, we implemented target generators using a single large sparse network and used masks to keep them identical to the original architecture from a logical standpoint. To achieve this, we extended PyTorch’s linear module (see GRouNdGAN’s API References: `src/layers/masked_linear.py`) to remove a neural network’s edges by masking its weights and gradients. The masked linear module accepts a binary mask matrix of dimensions $(N_{Input\ neurons}, N_{Output\ neurons})$ to initialize a fully connected layer. Setting element (n, m) in the mask matrix to zero removes the edge connecting input neuron n to output neuron m by zeroing out the corresponding weights and gradients. For the target generators, we defined the following three masks:

- input mask: Defines connections between the target genes’ generators and their regulating TFs and noise variables.
- hidden mask: Defines connections between the hidden layers such that there is no connection between hidden layers of two target genes’ generators.
- output mask: Defines connections between the hidden layers of each gene’s target generator and its outputted expression values (before the LSN layer).

The following pseudocode shows how we defined each mask matrix. In this pseudocode, the following notation is used:

- w : The width multiplier of target generator's hidden layers. As a result, the width of each target generator's hidden layer equals $w(N_{Noise} + N_{TFs_regulating})$.
- N_{Genes} : Number of target genes in the GRN.
- N_{TFs} : Number of regulating TFs in the GRN.
- N_{Edges} : Number of edges in the imposed GRN.
- N_{Noise} : Dimension of the noise variable per target generator.

The pseudocode for generating mask matrices

Start

```
// Compute the combined width (hidden dimension) of all target generators' hidden layers
combined (HDim) which equals  $HDim = w(N_{Noise} \times N_{Genes} + N_{Edges})$ . Each target generator's
hidden dimension is  $w(N_{Noise} + N_{regulating\_TFs})$ . If all genes were regulated by the same number
of TFs, HDim would equal  $N_{Genes}(w(N_{Noise} + N_{regulating\_TFs}))$ . Since that is not always the case,
the general formulation of HDim is equal to  $HDim = w(N_{Noise} \times N_{Genes} + N_{Edges})$ , where
 $N_{Edges} = \sum_{i=0}^{N_{Genes}} N_{regulating\_TFs\_i}$  and  $N_{regulating\_TFs\_i}$  is the number of TFs regulating  $Gene_i$ .
```

```
// Zero initialize mask matrices of shape (rows, columns)
```

```
Input_mask = zeroes( $N_{TFs}$ , HDim)
```

```
Hidden_mask = zeroes(HDim, HDim)
```

```
Output_mask = zeroes(HDim,  $N_{Genes}$ )
```

```
// This variable keeps track of the width of traversed hidden dimension
```

```
Beginning_HDim = 0
```

```
// loop through genes and for each gene, recover its index (Index_Gene) and the indices of its
regulating TFs (Index_regulating_TFs)
```

```
For (Index_Gene, Index_regulating_TFs) in causal_graph:
```

```
    // Calculate current gene's target generator width
```

```
    Current_HDim =  $w(N_{Noise} + N_{regulating\_TF\_indices})$ 
```

```
    // unmask neural connections between regulating TFs and the input layer of the gene's
target generator
```

```
    Input_mask[ Index_regulating_TFs, Beginning_HDim : Beginning_HDim +
```

```
    Current_HDim ] = 1
```

```
    // Initialize noise matrix
```

```
    // and unmask neural connections between noise inputs and the input layer of gene's
target generator
```

```
    Noise_mask = zeroes( $N_{Noise}$ , Hidden_Dimension)
```

```
    Noise_mask[:, Beginning_HDim : Beginning_HDim + Current_HDim ] = 1
```

```
    Input_mask = concatenate(Input_mask, Noise_mask) // vertically stacking (on rows
dimension)
```

```

// unmask neural connections between hidden layers of the gene's target generator
Hidden_mask[
Beginning_HDim : Beginning_HDim + Current_HDim,
Beginning_HDim : Beginning_HDim + Current_HDim
] = 1

// unmask neural connections between the last hidden layer and the output layer of the
gene's target generator
Output_mask[
Beginning_HDim : Beginning_HDim + Current_HDim,
Index_Gene
] = 1

Beginning_HDim += Current_HDim // Update width of traversed hidden dimension
End For
// Use the masks to initialize sparse linear layers to form the target generator.
End

```

We added the above description and pseudocode to the Supplementary Notes (File S1). To further clarify the pseudocode above, we have provided an example below. Consider a simple case where there are only three target genes regulated by two TFs, as shown in Figure R1-A. Here, we assume w and N_{noise} are both set to 2 and target generators have 3 hidden layers (It should be noted that GRouNdGAN was trained with the hyperparameter N_{noise} set to 1, but here we show the example with this value set to 2, to show that the concept/pseudocode works with different hyperparameter values). If one follows the steps of the pseudocode, the weight matrices shown in Figure R2 can be achieved, which when applied to a large neural network, results in the same neural network of Figure R1-B from a logical standpoint.

Figure R1. Logical implementation of the target generator (Panel B) corresponding to the GRN in panel A. In this example, the hyperparameters w and N_{noise} are both set to 2 and target generators have 3 hidden layers. It should be noted that GRouNdGAN was trained with the hyperparameter N_{noise} set to 1.

Figure R2. Mask matrices corresponding to the example GRN and its target generator in Figure R1. White and purple parts of the three binary matrices are set to 0 and 1, respectively.

Comment: “2. The supplementary table notations lack clarity. The authors provided three Excel files corresponding to Supplementary Table S4-S6. However, within each file, there are multiple tabs, and the associations are not well-defined. The Tables S7 and S8, mentioned in the paper, are also elusive and cannot be located.”

Response: Thank you for pointing this out to us. To facilitate interpretation of supplementary tables S4-S6 that are provided as separate xlsx files, we added a description sheet to each file to describe what each sheet shows. In addition, each sheet has its own legend/caption that describes the contents. Moreover, in the main text, we are now referring to specific sheets of these tables to enhance clarity.

Regarding Tables S7-S8: we had placed them in File S1 (along with Tables S1-S3 and captions of Tables S4-S6); however, we had missed to mention their location on page 45 of the main manuscript file in the section “Supplementary Material”, an oversight that now thanks to the reviewer’s comment is rectified.

Comment: “3. In 'Performance w.r.t. Resemblance' table, the paper asserts that GRouNdGan's performance on BoneMarrow is comparable with other datasets, yet I observed a RF AUROC of 0.75 on the test data, indicating subpar performance. Also, could you please provide insight into the substantial performance difference of GRouNdGan between its training and testing data for the BoneMarrow dataset?”

Response: This is indeed a very important point brought up by the reviewer and requires answer (and evaluation) from multiple perspectives. Before we provide that however, we want to clarify that we only mentioned in the manuscript that the realism of GrouNdGAN outperforms other simulators on this dataset, which is indeed the case: “Finally, GRouNdGAN outperformed all other simulators based on all metrics when we repeated the analysis using the BoneMarrow dataset, containing continuous cell states and much fewer cells”.

The first consideration here is the interpretation of resemblance scores for training set vs. resemblance scores for testing set. Due to the generative nature of the model, the interpretation of training resemblance and testing resemblance is quite different from that of supervised learning (i.e., predictive) models. For a generative model (such as GrouNdGAN), the resemblance of generated samples and training set is quite meaningful (unlike performance on training set for predictive models), since a good resemblance performance implies that the generative model has been successful in its main task of generating realistic samples. If the training resemblance performance is low, it means that it has failed at generating realistic samples. Now, when it comes to resemblance performance on testing set, a lower performance does not necessarily mean that the generative model has failed at its task. For example, if the test samples are not representative of the training set, a lower performance will be obtained, simply due to this issue. This is different from the case of a predictive model, since there, the goal is indeed making predictions on unseen samples, instead of simulating data similar to a training set (the goal of generative models). This however does not mean that one should only focus on training set resemblance metrics and completely ignore a test set (even though in many studies of generative models this is done). The reason we believe the test resemblance performance is still quite useful in generative models is that a model that simply copies the training set examples will have a very good training set resemblance score. That is why we provided both training set and testing set resemblance scores, which we believe should be considered simultaneously.

Comparison of testing set performance of GrouNdGAN vs. other methods shows that the worsening of the testing performance on the BoneMarrow dataset is not unique to GrouNdGAN, and in fact, GrouNdGAN

has a better performance compared to other simulators. The difference between training and test sets are usually more severe in small datasets (as a small dataset is more prone to sampling bias). So as our first hypothesis, we evaluated whether there is an obvious difference between the ratio of cell types in the real testing set and real training set. However, we could not readily identify any major difference between the ratio of cell types in the real test set and real training set.

Another explanation for the training/test performance difference could be the small number of training samples in the BoneMarrow dataset: a small training set means that the model will have a harder time learning from the data and some nuances will be lost during training. Consequently, we hypothesized that the main reason for this difference in testing/training performance is the number of samples in the dataset, affecting the ability of all simulators to learn from the data and also introducing (hard-to-detect) biases causing differences between testing and training set.

To test this hypothesis, we identified another hematopoiesis dataset (which we called Dahlin dataset), but with a much larger number of cells (n = 44,802 cells). When we simulated data using this much larger dataset, the resemblance performance difference between training and testing set significantly reduced (see Table R1 below, which is also provided in the new Table 3 of the manuscript).

Table R1: Resemblance performance on Dahlin dataset (n = 44,802).

	Training (GrouNdGAN)	Testing (GrouNdGAN)	Training (scGAN)	Testing (scGAN)
RF AUROC	0.52	0.54	0.56	0.57
Cosine distance	0.00011	0.00012	0.00021	0.00027
Euclidean distance	51.0	53.4	70.5	79.4
mILSI	1.91	1.91	1.90	1.90
MMD	0.014	0.026	0.016	0.033

As can be seen in this table, the performance difference between training and testing sets have significantly reduced (both for GrouNdGAN, and scGAN as a baseline). We repeated these analyses with yet another dataset (corresponding to follicular lymphoma tumour cells) with n = 89203 cells. Once again, a small difference between testing and training set performance was observed (Table R2 and 3 in the manuscript).

Table R2: Resemblance performance on tumour dataset (n =89203).

	Training (GrouNdGAN)	Testing (GrouNdGAN)	Training (scGAN)	Testing (scGAN)
RF AUROC	0.53	0.57	0.53	0.56
Cosine distance	0.00059	0.00098	0.00128	0.00066
Euclidean distance	133.8	173.9	198.7	141.4
mILSI	1.90	1.89	1.90	1.90
MMD	0.014	0.015	0.015	0.015

Based on these results, we believe the reason we observed a drop in the testing resemblance performance in the BoneMarrow dataset was its small number of cells. However, in spite of this, we observed that important biological characteristics of hematopoiesis (such as trajectories and pseudo-time) was captured by GrouNdGAN in the BoneMarrow dataset (as reported in Figure 5).

We added these new results and a summary of the discussion above to the revised manuscript in the results (Section “GRouNdGAN achieves high performance on other datasets” and tables and figures referenced within) and Discussion.

Comment: “4. In data simulation on PBMC-ALL, is it understood correctly that identical GRNs were applied across distinct cell types? I'm uncertain whether this might influence the expression of cell-type marker genes. Could you elucidate how your experimental setup ensures the consistency of cell type annotations?”

Response: Yes, the GRN inferred using GRNBoost2 on all cell types of the PBMC-All dataset was used by GRouNdGAN to generate synthetic data. It should be noted that the GRouNdGAN's objective is two-fold: 1) producing realistic cells and 2) imposing an input GRN through the target generator's architecture and minimizing labeler/anti-labeler losses. We have demonstrated that the realistic data generation objective on the PBMC-All dataset is met (Table 2, Fig. S2, Table S4) even when the same GRN is imposed across cell types. We suspected that this would not have been possible if the expression of cell-type markers were not preserved in GRouNdGAN's simulated data.

To verify this more directly, we generated a synthetic dataset using GRouNdGAN of the same size as the PBMC-All training set (67579 cells). We trained a classifier on the annotated cell types of the experimental training set and used it to annotate the cell types of the synthetic datasets. Then, we generated two dot plots to visualize the expression patterns of cell type marker genes in each cell type (Figure R1 in this document and Figure S4 in File S1). As can be seen in this figure, a high degree of concordance between the expression patterns of marker genes can be observed between the experimental and synthetic data. We have provided the results pertaining to this analysis in the Supplementary Notes (File S1) and Figure S4.

We would also like to mention that gene regulation is indeed cell type and context specific. Our goal in the reported analysis in Table 2 was to show that even when cell type specific information (whether cell types or cell type specific GRNs) are not provided to GRouNdGAN, it can still generate realistic data. Since such cell type information may not be available for some applications, this minimizes user involvement when it comes to annotating cell types or identifying cell type specific GRNs.

We have plans to expand GRouNdGAN in the future and provide the option to generate specific cell types on demand. Conditioning GRouNdGAN on cell type would be similar to the approach taken by cscGAN and cWGAN, which we used as baselines in our paper. However, imposing cell-type-specific GRNs would require a different target generator architecture (following the cell-type-specific GRN) for each cell type, which falls outside the scope of this study. However, if one wants to impose different GRNs on different cell types present in the training data, one option would be to separately train GRouNdGAN on each cell type (using cell-type specific GRNs) and generate data one cell type at a time (similar to data we generated for PBMC-CTL and the new PBMC-NaiveT (which we used for stability analysis in Table S5)).

Figure R1: Preservation of cell-type markers with GrouNdGAN-generated data. Dot plots of panels A and B, corresponding to simulated and real cells, respectively, visualize the fraction of cells per cell type expressing a marker through dot size and the mean standardized marker expression through color intensity.

Comment: “5. Line 218, did the same experiments also conduct on PBMC-ALL? I'm curious whether the diversity of cell types might impact the reconstruction of GRNs.”

Response: Thank you for this suggestion. We performed new benchmarking analysis (Table S6 - Sheet 4) using PBMC-All. While the trends were generally consistent with those of PBMC-CTL and BoneMarrow, the ordering of GENIE3 and GRNBoost2 were swapped, which may be due to their ability in working with a dataset containing different cell types. We added these results to the Results section, but we also added

a paragraph to the discussion, pointing out that follow-up analyses are needed to fully assess the effect of different cell types (and even different cell type-specific GRNs) on the performance of these methods.

Comment: “6. I suggest to involve more tools for cell-type-specific and cell-specific gene regulatory network construction, such as CNNC, DeepDRIM, CeSpGRN, LocaTE et al.”

Response: Thank you for this suggestion. We performed new analysis with CeSpGRN, a cell-specific GRN inference method, and added it to our benchmarks (Fig 5G-5H, Table S6).

With regards to CNNC and DeepDRIM, both of these methods are supervised GRN inference methods, which makes them very different in nature to the methods we benchmarked in this study, and would require their own specific data preparation, data splitting for evaluation, and training. Since these methods need to be trained on a training set of cells and their underlying GRN edges (as their labels), providing such ground truth GRN to them will place them at an unfair advantage to other methods in our benchmark; on the other hand, providing them with erroneous ground truth GRNs may place them in an unfair disadvantage. Similar trade off also exists with regards to the number of cells provided to them: should one provide them additional cells to form their training set (and hence result in a larger dataset for them compared to other methods), or divide the dataset into training and testing. Moreover, data splitting strategies such as C1/C2/C3 splits (suggested by Park and Marcotte (Nat. Methods, 2012) for protein-protein interaction networks¹) is also required to ensure no data leakage is present between training and testing sets in such supervised models (a consideration that does not apply to other methods benchmarked here). As a result, benchmarking these methods would require generating specific datasets and evaluating these methods considering all possible combinations of choices, which requires its own dedicated study to fully analyze these methods without placing them in an unfair advantage or unfair disadvantage. However, such a study is beyond the scope of this manuscript, which was focused on introducing GrouNdGAN and showcasing its ability for GRN inference benchmarking. Moreover, we faced some issues with the software of these two methods, including their libraries, environments, and preprocessing tools, which were not provided in their github repositories, making running their code, setting up the environments, and solving dependency issues rather challenging. Due to these reasons, we did not include them in the benchmarks of this study, but we mentioned/cited them in the Discussion and mentioned the need for a comprehensive study to benchmark these and various other categories of GRN inference methods.

With regards to locaTE and CeSpGRN, both of which are cell-specific GRN inference methods, we decided to only add CeSpGRN to our benchmarks as an example of this class of methods and leave the analysis of locaTE to a future study (due to the reasons discussed below). Based on our understanding, locaTE requires a transition matrix P as input (among other inputs) that encodes inferred cellular dynamics as a Markov chain on the cell state manifold. In their manuscript, they suggest various methods to estimate P , but it seems that this step is not a part of the locaTE method itself and should be performed using other tools. This, however, will introduce another level of uncertainty to the results of our benchmarking that would not be present in other benchmarked methods. In other words, it is challenging to know whether the potential (good/bad) performance of this method would be due to our choice of the method for inferring P , or due to the method itself. Moreover, since this paper is not published yet, it is possible that the code itself and the preferred method for calculating P would change during the publication process. We felt that it would be unfair to the authors of this method to benchmark them using potentially an unfinalized version of their code.

As a result, we decided to postpone benchmarking of the three methods above that require evaluation from many different angles and require their unique class of datasets to a future study, dedicated to benchmarking different GRN inference methods. We added the following paragraph to the discussion to emphasize the importance of these two classes of GRN inference method:

“Due to the flexibility provided by GrouNdGAN, one can evaluate a variety of GRN inference methods including supervised methods^{2,3} or cell-specific GRN inference methods^{4,5}. We included CeSpGRN⁴ in our benchmarks, but a comprehensive study that evaluates this and other methods of this class from different perspectives is of great interest.”

Comment: “7. It is better to show the performance of GRouNdGan if the number of TFs exceeds 15.”

Response: Thank you for your suggestion. To address this comment, and assess the effect of number of TFs regulating each gene beyond $k=15$, we simulated data using PBMC-CTL with $k = 20, 25, \text{ and } 30$. We added the results to Supplementary Table S5 (sheet 3) and Supplementary File S1 (page 2-3), where we discuss the effect of different GRNs (including different number of TFs) on the performance. We observed that increasing the number of TFs regulating each gene from 15 to 20, 25, and 30, results in marginal improvement of the performance (while increasing the computational cost). For example, the RF-AUROC on the test set for $k=15$ was 0.54, while for 15, 20, and 25, it was equal to 0.53.

Minor comments

Comment: “1. The authors indicated the generation of five replicates, each containing 1000 cells, to assess the robustness of the simulated data. Could you provide specific details about how these replicates were utilized to create Figure 2? For instance, were average values employed?”

Response: The stability analysis is a separate analysis from the t-SNE plots (shown in Fig. 2). In a typical application of a simulator, the user generates a batch of cells (say 1000) and uses them for follow-up analysis or the specific use-case they have in mind. That is why we simply used the first batch of 1000 cells generated by GrouNdGAN after training was complete. This allows us to evaluate its performance in a typical use-case, instead of averaging or aggregating multiple runs/batches of cells. One downside of using multiple batches/runs and then averaging would be that a simulator that generates not very realistic samples in each run, in average (across multiple runs) may produce more realistic cells, and this would artificially inflate the performance metrics and make the t-SNE/UMAP plots look better than how the model is used in a typical use-case. For these reasons, we used the results generated by the model in one run to generate Figure 2.

However, we also wanted to make sure that the generated samples by GrouNdGAN are stable and generating different batches of cells or different splits of train/test sets still achieve high performance. To test this, we performed the following stability analysis (some were reported in the original paper, but most correspond to new analysis):

First, we extracted scRNA-seq profile of two cell types from Zheng et al. ⁶. The first cell type corresponded to CD8+ Cytotoxic T cells (PBMC-CTL) and the second corresponded to CD8+/CD45RA+ Naive T cells (PBMC-NaiveT). We chose these, since they had the largest and second largest number of cells (and hence a large training set) in this study with $n = 20773$ and $n = 16,666$, respectively.

We performed the following stability analysis on each of these two datasets (we used two cell types to make sure the conclusions are applicable beyond a single dataset).

Using the CD8+ cytotoxic T-cells dataset (which corresponds to PBMC-CTL), we performed the following analyses:

- 1- We trained GRouNdGAN on the reference dataset. Then, we simulated five separate batches, each containing 1000 distinct simulated cells. Then, we assessed the resemblance of each simulated dataset with samples corresponding to real cells from both the training set and a held-out test set. This analysis allows us to assess the variation in the performance of the model across different generated datasets, all simulated based on the same training set and same GRN (the GRN for all simulated cells were identical in this part of the analysis and was obtained using GRNBoost2, as described in the manuscript).
- 2- Then, we re-trained GRouNdGAN two extra times, using two different training/testing split. Each such re-training not only involved different cells present in the training and test sets, but also meant that the GRN used by the model was different in each run (since the training set was used to infer a scaffold GRN). Similar to the previous analysis, we compared how realistic the cells are against real cells from both the training set and test sets.
- 3- We then repeated all the analyses above, this time using CD8+/CD45RA+ Naive T cells (which we called PBMC-NaiveT).

Overall, the analyses above resulted in 30 distinct simulated datasets, each containing 1000 cells, corresponding to two cell types, six distinct training/testing data and GRN and six trained models, each generating 5 distinct datasets.

Supplementary Table S5 (sheet 2, titled Stability Analysis) contains the results. Moreover, we added the description of these analyses to Supplementary File S1 (in Section titled “The stability of GRouNdGAN and the effect of GRN properties on its performance”). As can be observed in these tables, the results were quite stable and only a small degree of variability was observed for different generated batches, different training/test splits (which resulted in different GRNs). This was consistent for both cell types used in these analyses. For example, across all 30 different generated datasets, the miLSI comparing the simulated and held-out test sets ranged between 1.88 to 1.90, with a total mean and standard deviation of 1.89 (± 0.01). A similar degree of stability was observed based on other metrics using both training or test sets.

Comment: “2. It might enhance clarity to include the exact formulations of the cost functions for both the Labeler and anti-labeler, rather than a general mention of “minimizing the squared L2 norm.”

Response: Thank you for this suggestion. We added the following to Methods:

“More specifically, the corresponding loss for a batch of size N_{batch} is of the form $\frac{1}{N_{batch}} \sum_{i=1}^{N_{batch}} \|\hat{\mathbf{y}}_i - \mathbf{y}_i\|_2^2$, where $\hat{\mathbf{y}}_i$ is the estimated vector of TF expression values generated by the labeler or anti-labeler. For anti-labeler, \mathbf{y}_i corresponds to the TF expression values outputted by the causal controller; for labeler, this vector can also correspond to TF expression values from the real training data.

Comment: “3. In Figure 1, it appears that the input for the Critic module should be a vector rather than an entire matrix.”

Response: Thank you for pointing out this issue in how we had visualized the input to the critic. We re-made the Figure, and the updated version should clarify this aspect by adding vectors being inputted to the critic in Figure 1B and also 1C.

Comment: “4. In Line 150, when referring to "control," does this term imply the actual cells within the training/test sets? Furthermore, could you elaborate on the interpretation of the values in the row labeled "control" in Table 1?”

Response: The idea behind using “control” was that for various metrics (especially those for which the best score is equal to 0 such as MMD, Euclidean distance, Cosine distance), it becomes challenging to interpret very small numbers. While one can compare among different simulators to tell which one is better, there is not an easy/intuitive way to tell how good such small values are in the absolute sense. For example, while we know a small Euclidean distance is desirable between the simulated and real data, what constitute “small”? Should the values be between 10-100 to say they are small, or should they be between 0.01 to 0.1? To allow calibration of these scores, we included “control” metrics (described below).

For RF-AUROC, we simply used 0.5, since that was the best achievable performance (equivalent to the performance of a random classifier). But for less intuitive metrics (i.e., the other four), we calculated the “control” metrics as follows. For the training set (in Table S4 and Table 3), we randomly selected two disjoint sets of real training cells of the same size as the number of simulated cells and calculated the metrics between those two. We did not use all the training set to remove the number of cells as a possible confounding factor. For the testing set, we randomly divided the test set (aka the real experimentally measured cells) into two halves and calculated these metrics between the two. These control metrics allow us to calibrate the performance measures of simulators, because they tell us what degree of variation exists withing two randomly selected subset of real data (training or testing).

So to interpret the “control” values in Table 1, we use an example. The MMD between two halves of the real cells in the test set is equal to 0.012. Now looking at the same value for SPARSIM, we see that the MMD between the simulated data and test data for that is one order of magnitude larger than the control, while the performance of GrouNdGAN and scGAN is in the same order of magnitude. We were hoping that by including this control, the reader would be able to better interpret and calibrate the scores in this table.

Code and Software Review

“The author releases the source code of GRounNdGAN on Github and provides the guide to install and use it (<https://emad-combine-lab.github.io/GRounNdGAN/index.html>) with the demo dataset. I believe the installation using Docker and Singularity approaches should be okay (not tested), but it is not easy to run the program on HPC without root access.

I had some issues when installing GRounNdGAN:

1. Although the authors suggest users to utilize `Pyenv` to manage Python versions, it is only designed for managing multiple Python versions on the same node. It should be coupled with the plugin `Pyenv-virtualenv` (<https://github.com/pyenv/pyenv-virtualenv>) to allow users to keep code environments separately. It makes my packages from Conda crashed if I install GRounNdGAN following the instructions.
2. In addition, `source venv/bin/activate` can activate the created environment, but the installation path is still my user Python path.
3. When I ran `pip install -r requirements.txt`, the packages were installed into my python user path rather than the required path.

I have tried to fix the problems by myself using the steps below:

1. Install Pyenv (<https://github.com/pyenv/pyenv#basic-github-checkout>) and set the environment variable by hand; Install `Pyenv-virtualenv` (<https://github.com/pyenv/pyenv-virtualenv>)
 2. Run `pyenv install 3.9.6` to get a specific python version; Run `pyenv virtualenv 3.9.6 venv` to create the environment; Run `pyenv activate venv` to activate the environment; Run `echo \$PATH`, `which pip` and `which python` to check the path, pip tool, and python interpreter, respectively.
 3. Run `pip install -r requirements.txt`
 4. ModuleNotFoundError: No module named `packaging` occurred when installing `docopt==0.6.2`. Manually install it to solve the error: Run `pip install packaging --target ~/.pyenv/versions/3.9.6/envs/venv/lib/python3.9/site-packages`
 5. AttributeError: module `numpy` has no attribute `version` occurred when installing `statsmodels==0.11.0rc2`; Manually install it to solve the error: Run `pip install numpy --target ~/.pyenv/versions/3.9.6/envs/venv/lib/python3.9/site-packages`
- All required packages can be downloaded successfully, but I encountered other problems:
1. ERROR: Could not build wheels for docopt, pynndescent, session -info, sinfo, statsmodels, termcolor, tfrecord, umap-learn, which is required to install pyproject.toml-based projects. Manually install the setuptools and wheel packages: Run `pip install setuptools wheel --target ~/.pyenv/versions/3.9.6/envs/venv/lib/python3.9/site-packages --upgrade`
 2. ModuleNotFoundError: No module named `_ctypes`.
To solve this error, the `libffi-dev` is required. And, it needs root access, going back to the beginning. I did not successfully install GRounNdGAN eventually.”

Response: Thank you for introducing us to the pyenv-virtualenv plugin for pyenv. We have now added a link to pyenv-virtualenv’s installation guide and tutorial and suggested users to manage their environment using pyenv-virtualenv if they opt to use pyenv. Furthermore, we added a section to the local installation guide called “Troubleshooting”, where we list required packages for installation on Ubuntu. We tested Ubuntu and Windows installation on fresh installs of both operating systems in virtual machines. After installing the suggested packages, GRounNdGAN was installed without issues. Most if not all of these packages should be already available on HPC clusters, as they are required by core python packages.

Reviewer #2 (Remarks to the Author):

Comment: “I co-reviewed this manuscript with one of the reviewers who provided the listed reports as part of the Nature Communications initiative to facilitate training in peer review and appropriate recognition for co-reviewers.”

Response: Thank you. We hope that our responses have now addressed your comments.

Reviewer #3 (Remarks to the Author):

Comment: “Gene regulatory network (GRN) inference is crucial in many aspects of biological research, and the key challenge for this task is to obtain gold standard ground truth GRNs as well as the corresponding realistic dataset. In the present manuscript, the authors aim to generate scRNA-seq data based on known GRN. This is a computational biology problem of importance, and a successful framework will have widely application scenario.

However, current version of GRouNdGAN is not proved to be as powerful as described in the Abstract. More experiments and descriptions are required.”

Response: Thank you for your insightful comments. We have significantly revised the manuscript and have added various new analyses. We believe that thanks to the reviewers’ comments, the revised manuscript has now improved significantly, and we hope that the reviewers also agree with us.

General comments:

Comment: “1. Details of GRouNdGAN architecture and training procedure.

a). Please provide more details for pre-training in Fig. 1A. In the Critic module, how to calculate the Wasserstein Distance between the real scRNA-seq data and the generated expression vectors of TFs and genes? In addition, in my opinion, the Causal Controller learned the distribution of scRNA-seq data but not regulatory relationships between TFs and genes (which I will discuss below). Thus, if the pre-training scRNA-seq dataset contain different cell types, that is containing different data distribution, it is possible that the pre-train model will not converge.”

Response: We thank the reviewer for pointing out that we have not clearly explained the training and pre-training procedure, potentially causing misunderstanding for the reader. Below, we have clarified various important points regarding the training and pre-training, a summary of which we have also added to the manuscript (Results and Methods), which we believe will address the reviewer’s concerns. Moreover, we have now added an extra panel to Figure 1 (Figure 1A) that clarifies the flowchart of the training procedure and the overall architecture of each step. (We have also provided a discussion in response to Comments 2a and 2b, which is quite related to this comment).

The following points can clarify the reviewer’s concerns:

- 1) The data used to pre-train the causal controller (aka the NN with three hidden layers in Figs 1B and 1C) is identical between the two steps of the training of GRouNdGAN. We believe our use of the term “pre-training” has caused some degree of misunderstanding: in many applications, a NN is pretrained on a large dataset and then is fine-tuned on a different dataset of interest. For example, in image processing, various pre-trained models exist that one can fine-tune on a new application. In our manuscript, however, the pre-training is performed on the exact same training data on which the training of GRouNdGAN is performed (i.e., the two models in Fig. 1B and 1C are trained using the exact same set of reference real scRNA-seq data). As a result, the reviewer’s concern with regards to the “difference between cell types of pre-training data and training data” is not applicable here, as they are the exact same dataset. We have clarified this on page 6 of the revised manuscript: “The two steps of training above are performed on the same reference experimental scRNA-seq training dataset.”
- 2) Moreover, the pre-training (and the training) is repeated for each new dataset. For example, when using PBMC-CTL dataset, we would first pre-train the causal controller on the training set of PBMC-CTL (Fig. 1B). We then use the same PBMC-CTL training set to train the GRouNdGAN architecture in step 2 (Fig. 1C). On the other hand, when we use the BoneMarrow (or any other dataset), both

steps depicted in Fig. 1 (both pretraining and final training) must be redone using the same training data.

- 3) The causal controller (aka the NN with three hidden layers in Figs 1B and 1C) is tasked with generating realistic single cell TF expression values and not learning TF-gene relationships of a GRN: the imposing of TF-gene relationships is done in step 2 of training (depicted in Fig. 1C). The pretraining step (Fig. 1B) enables the causal controller to learn scRNA-seq data distribution (as correctly pointed out by the reviewer) and generate realistic simulated scRNAseq data pertaining to both genes and TFs of the training dataset, irrespective of any GRN. This is similar to scGAN, in which no GRN is considered, and a simulator is trained to generate realistic samples (irrespective of any GRN). After we pre-train the causal controller in Fig. 1B, its weights are frozen and is used to generate only the TF expressions. Even though the causal controller can generate expression values of target genes as well, we discard such values in step 2 (Fig. 1C), and only use the generated TF expression values. The reason is that (as the reviewer has correctly pointed out), the causal controller has learned the distribution of TF expressions of scRNA-seq, but not the regulatory relationships between TFs and genes in the user-provided GRN. In fact, our analysis reported in Fig. 3, corresponding to scGAN (orange dashed lines) confirms this and shows that if one simply uses the causal controller to generate both TF expressions and target gene expressions (as is done by scGAN and not by GrouNdGAN), then many TF-gene interactions that existed in the real data will be missed. Moreover, since no GRN is used in step 1 of training (aka pre-training in Fig. 1B), no relationship from the input GRN is imposed. This is why we only use the causal controller to generate realistic TF expression values (and step 2 of training is used to impose the GRN by the rest of the architecture, and not by the causal controller).
- 4) After the causal controller is trained to generate realistic TF expression values using the training set, an architecture depicted in Fig. 1C is used to impose (and not learn) the TF-gene interactions. This is a similar approach to that proposed by CausalGAN (and various theoretical and mathematical discussions are provided in⁷). Intuitively, the unique target generator NN of each target gene receives as input the TF expression values provided by the causal controller (corresponding only to its regulating TFs) and a noise value. Then, after going through the GrouNdGAN architecture (which includes a critic, a labeler and an anti-labeler), the model learns to generate realistic scRNAseq data such that the input TF-gene relationship provided as an input GRN is imposed.

In summary, as correctly pointed out by the reviewer, the causal controller learns the scRNA-seq data distribution of the training set in step 1 of training (aka pretraining of Fig 1B). Then, the causal controller is used to only generate TF expression values which are then inputted into target generators used to generate target gene expressions following the topology of a causal GRN. The same training dataset is used in both steps of training to train the model. This has multiple implications: for example, 1) we will not have any issues regarding discrepancy between the biological properties of training data in the two steps of training (as they are identical) and 2) the model cannot (and should not) be used to “predict” scRNAseq data in a new dataset or a new context and the model must be retrained on the new dataset (we have provided more detailed discussion regarding this point in response to comment 1b).

We have added the following paragraph to page 6 to summarize some of these points:

“The two steps of training above are performed on the same reference experimental scRNA-seq training dataset. The goal of the first step (pre-training, Figure 1B) is to train the causal controller to learn scRNA-seq data distribution and generate realistic data, irrespective of any GRN. The GRN edges between a set of regulators (e.g., TFs) and their targets are imposed in the second step of training (Figure 1C). To achieve this, in the second step of training, only the expression of regulators (TFs) outputted by the causal

controller are used and its output of target genes' expressions are discarded and instead are re-generated using the target generator neural networks in Figure 1C to enable imposition of causal relationships."

We also added the following to page 7:

"The combination of the model architecture (in that each target generator only receives a noise value and the simulated expression of its regulating TFs in the GRN) and the labeler/anti-labeler ensure that the GRN edges are causally imposed in the simulated data."

Moreover, we added a paragraph discussing these points to Methods (the excerpt of which is provided in response to the next comment).

With regards to your question about the Wasserstein distance, the following procedure is used. During training, and in each training step, the target generators generate a batch of simulated cells. These along with a randomly selected batch of real cells serve as inputs to the critic (implementing a discriminative function f_c), which in turn produces a real-valued score. The training objective of the critic involves maximizing the difference between the average score assigned to real ($\mathbb{E}_{x \sim \mathbb{P}_r} f_c(x)$) and generated samples ($\mathbb{E}_{x \sim f_g(\mathbb{P}_{noise})} f_c(x)$) with respect to its parameters following Equation 1 below. On the contrary, the generator attempts to minimize the average score the critic assigns to real and generated samples. Through this adversarial game, both the critic and generator co-evolve in a symbiotic manner and the generator learns a mapping f_g from a simple standard Gaussian distribution (\mathbb{P}_{noise}) to a distribution $f_g(\mathbb{P}_{noise})$ that approximates the real data distribution \mathbb{P}_r . An important point to consider is that the critic does not directly compute the Wasserstein distance (mathematically defined in Supplementary Notes - Details regarding the Wasserstein distance). Instead, it is encouraged to provide a meaningful estimate of the Wasserstein distance between the distribution of real data (\mathbb{P}_r) and the distribution of generated data ($f_g(\mathbb{P}_{noise})$) through its training process.

$$\min_{f_g} \max_{\|f_c\|_L \leq 1} \mathbb{E}_{x \sim \mathbb{P}_r} f_c(x) - \mathbb{E}_{x \sim f_g(\mathbb{P}_{noise})} f_c(x), \quad (1)$$

We added these details to the Methods section of the manuscript.

Comment: "b). As described in Methods, the imposed GRN is a form of bipartite directed acyclic graph (DAG). In this DAG, the number of genes is fixed to 1000 based on the most highly variable genes, and TFs are identified from these variable genes based on the AnimalTFDB3.0 database. GRouNdGAN trained Target generators for each TF. However, highly variable genes are different across different datasets, resulting in the dynamic TFs in DAGs. How to deal with the dynamic TFs while GRouNdGAN contains fixed Target generators or will GRouNdGAN train different Target generators for new dataset?"

Response: Thank you for this comment. For each experimental dataset, GrouNdGAN must be re-trained using experimentally measured scRNAseq and a GRN specific to that dataset. In other words, the GRNs (and hence the identity of considered TFs and their connections to the target genes) will vary from one dataset to another and different target generators need to be trained for a new dataset.

It is important to point out that GrouNdGAN is a reference-based simulator that as input accepts a user-defined GRN and a reference experimental dataset. Its goal is **not** to be trained on one dataset and then be a universal simulator for all datasets. As you have correctly implied, GRNs, highly variable genes, and other implicit and explicit biological properties are context-specific and dataset-specific. The goal of

GrouNdGAN is instead to 1) generate simulated scRNAseq data that resembles a specific reference dataset, while (most importantly) at the same time 2) impose an input GRN causally. A key consideration here is that the input GRN does not need to be the true causal GRN of the reference dataset (more on this later); as long as the GRN is consistent with the data (even at the correlation level), GrouNdGAN ensures that the input GRN is the causal underlying GRN of the simulated data (but not the reference data necessarily), while generating realistic simulated cells. This allows it to be used for development and benchmarking GRN inference methods, a process which requires the knowledge of the context- and dataset-specific underlying GRN for evaluation. Put differently, GrouNdGAN is a generative model and is not a predictive model in the sense that it does not try to predict scRNA-seq data for a different dataset.

We should also reiterate that the two steps of GrouNdGAN's training (pre-training in Fig. 1B and the full model in Fig. 1C) are done on the exact same reference training dataset using the same set of TFs and same set of target genes. This is different from the idea of transfer learning, in which one pre-trains a model on a large dataset of related (but not identical) training samples and then fine tunes the model. In GrouNdGAN, pre-training serves a different purpose: it is simply there to generate realistic TF expression values. Even though expression values of genes are also generated, they are discarded when we move to the second step and only TF expressions are used. To clarify this, we added the following to Methods: "We would like to point out that both steps of training (i.e., the pretraining of Fig. 1B and training of Fig. 1C) are performed on the exact same training set, and when data corresponding to a new dataset is to be generated, these steps need to be repeated. The goal of the first step is to train the causal controller to learn the distribution of training set and generate realistic TF expression values (without imposing TF-gene relationships). The goal of the second step is to use the TF expression values generated by the trained causal controller to generate expression of target genes while imposing the TF-gene causal relationships."

We also added a paragraph to Results, which we described in the previous comment.

Comment: "c). As described in Discussion, the authors pointed out that "The current version of GRouNdGAN only allows imposing a bipartite GRN and does not support regulation of TFs by other TFs". So, in DAGs, if the 1000 highly variable genes contain TF genes, will TF genes be discarded?"

Response: Thank you for pointing out that this sentence, as we had originally written it, is causing confusion. In the revised manuscript, we changed the sentence to "The current version of GRouNdGAN only allows imposing a bipartite GRN and does not support a multi-layer GRN, an assumption that we plan to relax with the future versions of the model" to be more accurate about what we intended to say.

With regards to your question, to avoid circular causality issues (i.e., a regulator A regulates target B, which itself regulates A again, forming a cycle), we have restricted the causal graph to be a bipartite TF→gene DAG. This is a common practice in the GRN inference literature and many GRN inference methods rely on this assumption. Additionally, the GRN at this point is in the form of a bipartite graph and the current version of the model does not support the graph of the form $A \rightarrow B \rightarrow C$. This is mainly due to its computational cost, and not due to any major or fundamental limit: we have performed preliminary tests with a 3 layer GRN and we did not observe any issue in generating data. Now, regarding your question, as long as GrouNdGAN is provided with a directed bipartite graph containing two **disjoint** sets of regulators and targets, it can work seamlessly. In other words, as long as a TF is not regulating any targets in the bipartite graph, it can be itself considered a target and its regulation by other regulators be included in the bipartite graph (as long as the graph remain a DAG). However, in the analyses reported in the manuscript, we used the TFs that were among the highly variable genes to be in the "regulator" set with the potential to regulate the genes in the "target" set.

Comment: “d). In line 168, “generate samples from different cell types” means generate scRNA-seq data containing different cell types or generate scRNA-seq data of a different cell type which is not appeared in training set? If cell type information is available a priori, how to import this priori knowledge into GRouNdGAN?”

Response: We would like to thank the reviewer for identifying this point of confusion for the reader. Reading over this sentence, we now realize that it can be read in two different ways. To clarify this, we replaced that sentence with the following few sentences:

“One of the challenges faced by GANs is the possibility of mode collapse: when applied to heterogenous datasets, they may learn to generate only a limited set of examples, instead of generating a variety of examples from the entire distribution of the training data. We trained GrouNdGAN using PBMC-All, which contains multiple cell types, to determine whether it can generate samples from different cell types present in this dataset, or it will suffer from mode collapse and only generates samples corresponding to the most represented cell type. This is particularly important, since many scRNA-seq datasets are heterogenous and contain different cell types or cells in different states.”

So to clarify, “generate samples from different cell types” was referring to generating simulated cells from cell types present in the training set, and not out of distribution generation (which is a different problem and not the goal of this study).

Now if cell type information is available, different strategies can be used. If the user wants to impose a single GRN onto the generated samples (independent of their type), no special modification is needed, and they can directly use GrouNdGAN for the full heterogenous training data (as we showed with PBMC-ALL analysis or with BoneMarrow dataset). If different GRNs are to be imposed for each cell type, the current strategy would be to divide the training (experimental dataset) into different cell types and generate simulated cells separately for each cell type (similar to what we did using PBMC-CTL and the new PBMC-NaiveT subsets of the PBMC-ALL dataset). In Discussion, we have mentioned “We should note that it is trivially possible to use cell type/cluster information with GRouNdGAN: one can simply provide subsets of the experimental dataset corresponding to a cell type/cluster as a reference, one at a time, with a cell-type specific (or shared) GRN (similar to our analyses using PBMC-CTL and PBMC-NaiveT in Table S5-Sheet 2).”

We should also add that, as we have discussed in the Discussion, one extension of this model that we are considering is to use a causal conditional GAN framework for this purpose: “Additionally, a conditional version of GRouNdGAN can be developed, allowing the user to generate cells of specific cell type or on a specific stage along a lineage trajectory” on page 35. However, this is one of our plans for the future and currently the above two strategies are supported.

Comment: “e). How exactly did GRouNdGAN solve the challenge that when cell type information is unavailable? The authors only described the results on two datasets (human PBMC and mouse BoneMarrow) but did not provide the details how they performed.”

Response: We are not sure if we fully understand this comment regarding missing details of results, but since the first reviewer also had difficulty finding some of the supplementary material, we suspect that it stems from the same issue, which we have tried to resolve in the revised manuscript by providing the exact location (including the sheet number within excel files) of each table and also by providing

description sheets to the supplementary excel files. We would like to emphasize that we had provided many details of the performance of the simulated data from many different perspectives on PBMC-CTL, PBMC-All, BoneMarrow, and have now added additional results corresponding to PBMC-NaiveT dataset (a subset of PBMC-All that we used for stability analysis), Dahlin dataset (an independent dataset corresponding to mouse bone marrow haematopoietic stem and progenitor cells differentiation towards different lineages), Tumour-All dataset (containing cancer cells and immune cells from tumour biopsies), and Tumor-malignant dataset (containing only the malignant cancer cells of the previous dataset). We have analyzed the performance of GrouNdGAN on these datasets from many different perspectives, described and discussed in main tables/figures and supplementary tables/figures (9 supplementary tables and 26 supplementary figures). In addition to the main figures and tables that are provided in the main manuscript file, all supplementary figures are provided in File S1 (in addition to some more details about the results and methods, which are described in Supplementary Notes section of this file). Additionally, all supplementary tables that fit in one page are provided in File S1. Three supplementary tables, each with several sheets, that contain the bulk of the details of the results are provided as separate excel files (Table S4, Table S5, Table S6). In the revised manuscript, and to make finding the results of each analysis easier in these tables, we have added the sheet number that contains the tables for each particular analysis. If there is any specific detail that the reviewer is referring to (and is not already available in these figures and tables), we would be happy to provide them.

With regards to how GrouNdGAN solves the challenge when cell type information is unavailable, we have shown that it can generate realistic cells, imposes GRNs, captures trajectories and behaviour of markers across pseudo-time axes. Since PBMC-ALL, BoneMarrow, Dahlin, and Tumour-All contain different cell types and cell states, but this information was not inputted to GrouNdGAN, the results corresponding to these datasets directly show that even in the absence of cell type information, GrouNdGAN can achieve its goals (e.g., generating realistic cells from different cell types, imposing GRN edges, ...).

Comment: “f). Figure 1 is rough. Neural networks diagrams are redundant while details for framework input/output and training process are inadequate. I suggest the authors discuss with other colleagues and redraw it.”

Response: Thank you for this suggestion. To address this, we added an extra panel to Figure 1 (i.e., Figure 1A), which shows the flowchart of the training procedure and also a high level view of the architecture of the neural networks in the two steps of training. Additionally, we changed some other details of Figures 1B and 1C to further clarify the inputs. Moreover, we added extra details regarding the training to the caption of this figure, and also to the text of the manuscript.

Comment: “2. Whether the framework learns regulatory relationships is debatable.

a). As discussed in Comment 1.a, the three-layer neural networks learn the gene expression distribution rather than regulatory relationships. There are inherent shortcomings to the assumption that gene expression patterns correspond to regulatory relationships (Aditya Pratapa et al., Nat. Meth., 2020; Mark W.E.J Fiers et al., Brief Funct Genomics, 2018; Hao Li et al., Nat. Mach. Intell., 2022).). To investigate the causal interactions between TFs and genes, multiple modalities data such as (sc)ATAC-seq, TF ChIP-seq are required. Thus, module description of Causal Controller in GRouNdGAN is not proper. This limitation for inferring GRN using scRNA-seq data should be discussed.”

Response: We thank the reviewer for pointing out a very important consideration in causal GRN inference. We would like to first start by mentioning that we agree with most points raised by the reviewer, and we have added a paragraph to the Discussion, pointing out that if one wants to impose the causal GRN of the

training data in the simulated data, multiple data modalities are needed to obtain more accurate GRNs. That being said, we would like to argue that access to the true underlying GRN of the real data, while necessary for some potential applications of GrouNdGAN in the future, is not necessary for the main goal of the current study (aka simulating datasets for GRN benchmarking):

- 1) We agree with the reviewer that simply learning the expression patterns of a dataset does not guarantee that the model has learned or has imposed TF-gene relationships. As we pointed out in response to Comment 1a, even our own reported analysis of scGAN in Fig. 3 supports this claim. In the corresponding analysis, we observed that when GRNBoost2 is applied to learn a GRN using scGAN generated data (that only implicitly learn TF-gene interactions by learning gene expression patterns/distributions), some interactions that could have been identified in the real data no longer are identifiable in the simulated data (see the analysis corresponding to Fig. 3 in the manuscript for more details). However, in GrouNdGAN, we are not simply relying on learning scRNA-seq data distribution (as is done by scGAN), but rather use the architecture of Fig. 1C to impose a causal GRN onto the data.
- 2) A very important consideration is that GrouNdGAN does not try to *learn* the *true* causal GRN of the data (more on this in the next points), nor it requires knowledge of it as input. In fact, GRNBoost2 is not a causal GRN inference method and quite likely, many edges identified by it correspond to correlation edges and not causation edges. However, when the “scaffold” GRN found by GRNBoost2 on the real training data (or some variation of it, as we used in various analyses), is used as input to GrouNdGAN, the GRN becomes the underlying causal GRN of the *simulated* data (and not of the training real data). To put another way, it is possible (and quite likely) that the underlying true causal GRN of the training data and the simulated data are different from each other. As correctly pointed out by the reviewer, finding the true causal GRN of the real dataset is not a trivial task and requires many different modalities (assuming that it would be even possible). To further emphasize this point, we performed a new analysis in which we learned the input GRN of GrouNdGAN using PPCOR (which is a purely correlation-based method). When performing TF knockout experiments (new Figure S5 in File S1) on the GrouNdGAN generated data, once again we observed that knockout of the TFs significantly changes the expression of their targets (in 75.2% of cases). Since the PPCOR-inferred GRN is distinct from the GRNBoost2-inferred GRN, this further emphasizes that the imposed GRN does not need to be the true underlying causal GRN of the real training data, for it to become the causal GRN of the simulated data. We added the new analysis to the section “GrouNdGAN imposes a causal GRN in the simulated data” in Results.
- 3) We would also like to add that we completely agree with the reviewer that gene expression alone is not sufficient to learn the true underlying causal GRN, and different modalities (such as scATAC-seq), TF ChIP-seq, etc. should be used to improve the GRN inference performance. However, we need to make two important points:
 - a. In GrouNdGAN, we are not trying to perform a GRN inference task (within the model itself), but rather we are aiming to impose an input GRN onto simulated data that is realistic. As such, one could use different data modalities to infer a more accurate GRN using the real data and impose that using GrouNdGAN, if desired. Alternatively, as we have shown here, even a GRN that is learned from scRNA-seq data alone (and may not correspond to the true underlying GRN of the real data) can be used with GrouNdGAN if the goal is to benchmark GRN inference methods (which was our main goal in developing GrouNdGAN).
 - b. In the context of using data generated by GrouNdGAN to benchmark a GRN inference method, we should point out that many methods have been developed to perform GRN inference using scRNA-seq alone (e.g., the various methods benchmarked in the BEELINE

study). While these methods may be relying on scRNA-seq alone (which as pointed out by the reviewer is a limited assumption), they have been able to find important biological processes in different contexts. Moreover, the wider availability of scRNA-seq data (compared to other single cell data modalities), make it useful for the community to develop methods that (while not perfect), can learn a GRN using scRNA-seq alone. For such methods, GrouNdGAN can be used for benchmarking and evaluation. Finally, even when multiple data modalities are used by GRN inference methods, scRNA-seq is often one of such modalities (e.g., CeSpGRN⁴) and GrouNdGAN-generated data can be used to assess the component of the multi-modal GRN inference method that uses scRNA-seq data. That being said, we believe that one impactful future direction would be to develop a multi-modal simulator that imposes a causal GRN in different data modalities (a direction that we are quite interested in, but falls out of the scope of this study).

Comment: “b). The authors showed the ability of GrouNdGAN in imposing the input causal GRN by performing in-silico TF knockout experiments. Further, the authors showed that imposed GRN can be reconstructed from the simulated data using existing methods. Above tests are not sufficient to demonstrate that GRN information is integrated the generated scRNA-seq data. As discussed in Comment 2.a, neural networks can learn the expression pattern from scRNA-seq data and are possible to be powerful to generate simulated data without any GRN information. Model training with completely random GRNs and with no GRN imposed are necessary to test this point.”

Response: We have performed additional analyses and below, we will discuss the results corresponding to data simulation with random GRN and no GRN (and also with GRN found by a correlation-based method). However, before explaining that, we should clarify several points, which we believe have caused some misunderstanding.

The reviewer mentioned that “As discussed in Comment 2.a, neural networks can learn the expression pattern from scRNA-seq data and are possible to be powerful to generate simulated data without any GRN information.” We completely agree with this point, and we are not arguing against that. We may be mistaken, but based on this and other comments, we believe the reviewer is under the impression that in this study, our goal was to generate simulated data that is very realistic, and to achieve this, we have used a GRN to assist with the task of generating realistic scRNAseq data (to refer to this goal, we call it Goal1, henceforth). However, this has not been our goal: generating realistic looking scRNAseq is already achieved by many simulators, including scGAN (which our model builds on). In fact, our various analyses have shown us that it is much easier to generate realistic looking scRNAseq data *without imposing any GRN*.

In this study, however, our goal was to generate a scRNAseq data on which a user-defined GRN is imposed, and the data is realistic and resembles the training set (despite the strict requirements of imposing a GRN) (to refer to this goal, we call it Goal2, henceforth). Had our goals was indeed Goal1, we would understand the concerns of the reviewer. In such a scenario, indeed one may be concerned that realistic looking scRNA-seq data can be generated irrespective of a GRN and then one can question the point of imposing a GRN. And we already know that if the goal is simply to achieve the best resemblance scores, one does not need to impose a GRN: scGAN achieved very high-quality data without imposing any GRN. Moreover, had our goal was Goal1, we would understand why the reviewer is concerned that the GRN we are imposing may not correspond to the true causal GRN of the training real dataset, and why one may need additional data modalities to find that (previous comment).

However, Goal1 is not our goal and instead our goal was Goal2. In fact, if it was possible (in an ideal theoretical sense), we would have liked a simulator that can impose any random GRN on the data and still ensure that the generated simulated data resembles a fixed training real dataset (even if that meant a high resemblance to real data, but not necessarily the highest). One can, however, easily argue that such an ideal case is impossible (which we have discussed in the manuscript on page 34 in Discussion and other places in the manuscript). To understand why, instead of causality, we can focus on a more intuitive and simpler-to-explain “correlation pattern”. Imposing a 1) correlation (or causation) pattern on a simulated data, while at the same time 2) requiring the simulated data to be indistinguishable from a training real dataset (which is a much stricter requirement than asking the simulated data to have a generally similar distributions such as negative binomial) can be contradictory to each other, if the correlation pattern to impose is inconsistent or contradictory to that of the real dataset. Because such correlation (or causation) patterns can be picked up by a classifier or in any of the other resemblance metrics to make the simulated data distinguishable from the real data. This is why we used the real training dataset to first find a “scaffold” GRN network that is consistent with the real data. However, one of the strengths of GrouNdGAN is that different variations of a GRN (that is consistent with the real dataset) can be used and can be imposed, while still achieving high resemblance to the real training dataset.

For example, in the analyses reported in section “The imposed GRN can be reconstructed from the simulated data” and the analyses reported in section “Benchmarking GRN inference methods using GrouNdGAN confirms prior insights from curated experimental datasets” we first applied GRNBoost2 to the real training dataset and identified top 10 TFs for each gene. But as input to the GRN, we did not provide the most consistent GRN to GrouNdGAN; instead, we provided (and imposed) a GRN in which the interaction between a gene and top identified 1st, 3rd, 5th, 7th, and 9th are imposed (imposed GRN), but the edges corresponding to top 2nd, 4th, 6th, 8th, 10th TFs are not imposed (unimposed GRN). We see that the imposed GRN can be picked up by different GRN inference methods (but with different degrees of accuracy, but consistent with BEELINE study and their biological benchmark; see Figure 5G and 5H). On the other hand, all methods performed close to random on the unimposed GRN, which was still important and consistent with the real training data (see Supplementary Figures S6-S8). We also swapped the role of imposed and unimposed edges and observed similar conclusions, as reported in Table S6 – Sheet 7.

The point we are trying to make here is that while GrouNdGAN requires a GRN relatively consistent with the training data (due to the contradictory nature of imposing an inconsistent correlation/causation pattern and making simulated data indistinguishable from real data), one still has some degree of freedom in varying and modifying the consistent GRN to enable generating different benchmarks. In fact, our analysis of imposing half of the top identified edges for each gene and not imposing the other half can be used such that the unimposed edges act as a negative control.

We should also add that in the Discussion, and also in Supplementary Notes (File S1) in section “The stability of GrouNdGAN and the effect of GRN properties on its performance”, we discuss the importance of the consistency of the GRN with the real training data. For example, we compared the scenario in which we imposed the top 10 TFs of each gene vs. the scenario in which we imposed the bottom 10 TFs (lowest ranked by GRNBoost2). Consistent with our expectation, we observed that when the model tries to impose a very inconsistent GRN, the data resemblance metrics deteriorate (due to its contradictory requirement).

With regards to your request for results on no GRN and random GRN:

GrouNdGAN becomes equivalent to scGAN when no GRN is imposed (aka the model and its training will only include the architecture shown in Figure 1B and the pretraining steps). We have already reported on the performance in such a scenario. As can be seen in Table 1-2, and Supplementary Table S4, without

imposing a GRN, one can obtain realistic looking simulated data achieving good resemblance metrics. However, the downside is that we have not achieved the main goal of GrouNdGAN: no causal GRN is imposed and the underlying data generating causal GRN is unknown (and hence the simulated data cannot be used for GRN inference benchmarking). Moreover, as Figure. 3 shows (comparing the orange dashed lines against green dotted lines), some of the interactions that were important in the original training set are missed.

In addition, we performed new analyses imposing two random GRNs and also GRNs inferred by PPCOR (a correlation-based method). When we simulated data in which the input to GrouNdGAN was a random GRN, as expected, the resemblance metrics were quite poor. For example, even on the training set, the miLSI was 1.17 (Table S6 - Sheet 8). We repeated that with another random GRN and once again, we observed poor performance (Table S6 - Sheet 8). These observations further confirm our previous observations that when the input GRN is inconsistent with the training data, imposing them results in a dataset that is easily distinguishable from the training real dataset. We have added these results to the section “The imposed GRN can be reconstructed from the simulated data”:

“Next, we repeated the analyses above on the PBMC-CTL dataset, but this time with input GRNs learned by PPCOR (a GRN inference based on partial correlation) from the training set. Similar to the results above, GRNBoost2 could reconstruct the imposed edges, and the importance of the imposed edges were accentuated by GRouNdGAN (compared to the original training dataset) (Table S6 – Sheet 8). We also imposed two randomly generated GRNs (with similar density to the GRNs above in which each gene was regulated by 5 TFs). Once again, the imposed edges were accentuated by GRouNdGAN, however imposing a random GRN (whose induced patterns may be inconsistent with real reference data) came at the cost of lower resemblance between simulated and reference data. (See File S1 and the Discussion for more details on why imposing a GRN that is inconsistent with the reference data (e.g., a random GRN) is a contradictory requirement to generating simulated data resembling the reference data).”

Comment: “3. Performance evaluation is insufficient.”

Response: Thank you for your suggestion for extra performance evaluations. We have included various additional analyses, as per your suggestions, which we believe has significantly improved the quality of the manuscript.

Comment: “a). Model stability. It is necessary to test the consistency of multiple generated scRNA-seq for the same cell type.”

Response: Thank you for this suggestion. To address this comment, we performed the following analyses. First, we extracted scRNA-seq profile of two cell types from Zheng et al. ⁶. The first cell type corresponded to CD8+ Cytotoxic T cells and the second corresponded to CD8+/CD45RA+ Naive T cells. We chose these, since they had the largest and second largest number of cells in this study with $n = 20,773$ and $n = 16,666$, respectively.

We performed the following stability analysis on each of these two datasets (we used two cell types to make sure the conclusions are applicable beyond a single dataset or a single cell type).

Using the CD8+ cytotoxic T-cells dataset (which corresponds to PBMC-CTL), we performed the following analyses:

- 1- We trained GRouNdGAN on the reference dataset. Then, we simulated five separate batches, each containing 1000 distinct sets of simulated cells. Then, we assessed the resemblance of each

simulated dataset with samples corresponding to real cells from both the training set and a held-out test set. This analysis allows us to assess the variation in the performance of the model across different generated datasets, all simulated based on the same training set and same GRN (the GRN for all simulated cells were identical in this part of the analysis and was obtained using GRNBoost2, as described in the manuscript).

- 2- Then, we re-trained GRouNdGAN two extra times, using two different training/testing split. Each such re-training not only involved different cells present in the training and test sets, but also meant that the GRN used by the model was different in each run (since the training set was used to infer a scaffold GRN). Similar to the previous analysis, we compared how realistic the cells are against real cells from both the training set and test sets.
- 3- We then repeated all the analyses above, this time using CD8+/CD45RA+ Naive T cells (which we called PBMC-NaiveT).

Overall, the analyses above resulted in 30 distinct simulated datasets, each containing 1000 cells, corresponding to two cell types, six distinct training/testing data and GRN and six trained models, each generating 5 distinct datasets.

Supplementary Table S5 (sheet 2, titled Stability Analysis) contains the results, and the description of these analyses are added to Supplementary Notes (in File S1 in Section titled “The stability of GRouNdGAN and the effect of GRN properties on its performance”). As can be observed in these tables, the results were quite stable and only a small degree of variability was observed for different generated batches, different training/test splits (which resulted in different GRNs). This was consistent for both cell types used in these analyses. For example, across all 30 different generated datasets, the miLISI comparing the simulated and held-out test sets ranged between 1.88 to 1.90, with a total mean and standard deviation of 1.89 (± 0.01). A similar degree of stability was observed based on other metrics using both training or test sets.

Comment: “b) The authors used three scRNA-seq datasets to train the model, and also used these datasets to test it. It is necessary to test the performance of GRouNdGAN on new datasets with GRNs information, which are not appeared in current training datasets. Specially, gene expression patterns as well as GRNs of tumor cells are often different from normal cells, the performance evaluation of GRouNdGAN on tumor data is needed.”

Response: Thank you for your suggestion to use additional datasets. In addition to the datasets discussed earlier, we applied GRouNdGAN to three other datasets (results provided in a new subsection of Results titled “GRouNdGAN achieves high performance on other datasets”). One corresponded to cells undergoing hematopoiesis (similar to the BoneMarrow dataset), but with much larger number of cells ($n = 44,802$)⁸ (which we called Dahlin dataset). The second dataset corresponded to malignant cells and cells present in the tumor microenvironment of 20 fresh biopsies from follicular lymphoma tumors ($n = 136,147$), which we called the Tumor-All dataset⁹. We also formed a dataset containing only the malignant cells present in this dataset ($n = 89,203$ cells), which we called the Tumor-malignant dataset. Table 3 and Table S4-Sheet 2 show the performance of GrouNdGAN in generating realistic scRNA-seq data, while Figures S17-S19 (in File S1) show the tSNE plots of simulated and real cells. For all three datasets, different metrics show that the generated data has a high degree of similarity to experimental data and there is a small difference between the training and test set performances.

We would however like to clarify an important point. GrouNdGAN is a generative model, and not a predictive model. As such, its use-case and goal is to take a training experimental dataset, impose a user-

defined **causal** GRN (that can come from the experimental dataset or can be a variation of that), and generate simulated data that 1) resembles the training set and 2) have encoded the causal GRN relationship among the TFs and genes. While we have divided each dataset into training and testing sets and trained the model on the training samples and evaluated on the testing set with regards to resemblance (e.g., Tables 1-2, Figure 2), the model cannot and should not be used to train on one dataset and predict scRNA-seq in another dataset. Such a setup is appropriate for predictive models and does not apply to the use-case or aims of GrouNdGAN. Below, we will detail why GrouNdGAN cannot and should not be used for such an application:

- 1- GrouNdGAN does not include a GRN inference module and the GRN is provided by the user. In our analyses, we used GRNBoost2 as one off-the-shelf method to learn a scaffold GRN from the training set, and we used different variations of the learned GRN to impose. In the new analyses, we also show that other methods (e.g., PPCOR which is simply a correlation-based method) can also be similarly used.

A “predictive” model that wants to predict scRNA-seq of an independent test dataset (coming from a different context), from a training dataset, would inevitably require the availability or inference of the *true causal underlying GRN governing both* datasets. This is something that is not within the scope of GrouNdGAN and in our use-cases, we do not expect availability or inference of such GRNs. In fact, we have pointed out that the input GRN does not need to be the true underlying causal GRN of the experimental training set (e.g., it could involve spurious correlations). However, when used with GrouNdGAN, the used GRN will become the true causal GRN of the simulated data (and not the experimental data). This is a very important distinction. In fact, in our new analyses, we used PPCOR, which is completely based on correlation (and not causation) and we again show that GrouNdGAN can be used in this setup.

Also, a “predictive” model that wants to train on one context and predict the scRNA-seq in another context (e.g., scGen¹⁰, not to be confused with scGAN which we used in this study), requires other assumptions and steps, as the GRN of the training dataset most likely will not be relevant to the testing context. None of these applications (while very important) were our goals or within the scope of this study.

- 2- GrouNdGAN does not even aim to predict the scRNAseq profile of cells in the testing set, but rather tries to simulate new cells similar to the training set, while imposing a causal GRN. Unlike non-generative (and predictive) machine learning algorithms in which the performance on the training set is completely unreliable and irrelevant (except for debugging), in a generative model resemblance of the simulated samples to the training set is quite meaningful and is the primary goal. However, if focusing on training set performance alone, one needs to be careful about the possibility that the generative model would simply repeat (copy) the training examples. That is why in our manuscript, we reported on both the training set resemblance performance and testing set performance (see for example Table 3 or Table S4).

To reiterate, the goal of generative models (including GrouNdGAN) is to generate samples that are similar to the training set (but potentially with some additional properties (like GrouNdGAN for GRN inference benchmarking) or without such additional properties (such as scGAN for data augmentation)). Unlike predictive models (e.g., scGen¹⁰) that try to predict on unseen (and in some cases out of distribution) samples, predicting on out-of-distribution datasets or contexts is not the goal for such generative models.

We would like to point out, however, that we absolutely agree with the reviewer that a predictive model developed and designed based on GrouNdGAN that can predict scRNAseq data in out-of-distribution cell types or contexts, could be an amazing contribution. We are working on such approaches, however as we mentioned earlier, this requires the ability to infer the true causal GRN underlying the experimental data.

Our early investigations towards this direction (which we have included in the manuscript in the TF knock out section) shows that this is a worthy direction to pursue, as even the GRN learned by GRNBoost2 (which is not a causal GRN inference method) captures some signals that are generalizable on unseen contexts. In the revised manuscript, we have updated the Discussion to clarify that predicting on new contexts, cell types or datasets is not the goal of GrouNdGAN, and given a new dataset, the model needs to be retrained (Page 34).

Comment: “c) Pseudo-time analysis and In-silico perturbation experiments are also needed to be conducted on another new dataset.”

Response: Thank you for this suggestion. We added new analyses corresponding to pseudo-time and trajectory inference from a new independent dataset (aka Dahlin dataset), and we performed in-silico perturbation analysis using a tumor dataset containing malignant cells as well as cells in the tumor microenvironment:

We obtained another hematopoietic dataset corresponding to the scRNA-seq (10x Genomics) profiles of 44802 mouse bone marrow haematopoietic stem and progenitor cells (HSPCs) differentiating towards different lineages from GEO (accession number: GSE107727). First, we generated simulated data based on this dataset and assessed the resemblance metrics. All metrics not only showed a high degree of resemblance between real and simulated data, but also showed the performance on the test set was very close to the performance on the training set (the new Table 3). Visually, t-SNE plots (shown in Figures S17) also confirmed the realism of simulated data, which we had quantified using the measures reported in Table 3 and Supplementary Table S4.

We then repeated the trajectory inference and pseudo-time analysis and evaluated the activation patterns of the erythrocyte, neutrophil, and monocyte (akin to the analysis reported in Figure 5 for the BoneMarrow dataset). Once again, we observed a high degree of concordance between the results obtained from the simulated data, and those obtained from the real experimental dataset (Figure S20).

We reported the results of this analysis on page 23-24 of the manuscript and in the Tables and figures mentioned in the above paragraphs.

Next, we obtained a scRNA-seq dataset corresponding to malignant cells and cells present in the tumor microenvironment of 20 fresh biopsies from follicular lymphoma tumors (n = 136,147), which we called the Tumor-All dataset. We also formed a dataset containing only the malignant cells present in this dataset (n = 89,203 cells), which we called the Tumor-malignant dataset. Table 3 and Table S4-Sheet 2 show the performance of GrouNdGAN in generating realistic scRNA-seq data, while Figures S18-S19 (in File S1) show the tSNE plots of simulated and real cells. For all three datasets, different metrics show that the generated data has a high degree of similarity to experimental data and there is a small difference between the training and test set performances.

We used the Tumor-All dataset to perform cell type specific TF knockout analysis on four cell types. The results obtained using this analysis confirmed the observations we had made using TF knockout analysis of PBMC-All dataset (Figure S26). We reported these results in section “In-silico perturbation experiments using GRouNdGAN” and the tables and figures referenced in the corresponding paragraph.

Minor Comments:

Comment: “1. Line 46, full name of DREAM.”

Response: We added the full name of DREAM challenges (Dialogue for Reverse Engineering Assessment and Methods) to the paper.

Comment: “2. Line 123, Typo “scRAN-seq”.”

Response: Thank you for spotting this typo. We changed “scRAN-seq” to “scRNA-seq”.

Comment: “3. Line 140, “red and blue” inconsistency between figures and texts.”

Response: We thank the reviewer for finding this inconsistency. We confirm that blue and red dots in Figure 2-A and 2-B correspond to real and simulated cells, respectively. We modified Figure 2's caption to reflect this fact.

References

- 1 Park, Y. & Marcotte, E. M. Flaws in evaluation schemes for pair-input computational predictions. *Nature methods* **9**, 1134-1136 (2012).
- 2 Chen, J. *et al.* DeepDRIM: a deep neural network to reconstruct cell-type-specific gene regulatory network using single-cell RNA-seq data. *Briefings in bioinformatics* **22**, bbab325 (2021).
- 3 Yuan, Y. & Bar-Joseph, Z. Deep learning for inferring gene relationships from single-cell expression data. *Proceedings of the National Academy of Sciences* **116**, 27151-27158 (2019).
- 4 Zhang, Z., Han, J., Song, L. & Zhang, X. Inferring cell-specific gene regulatory networks from single cell gene expression data. *bioRxiv*, 2022.2003. 2003.482887 (2022).
- 5 Zhang, S. Y. & Stumpf, M. P. Learning cell-specific networks from dynamical single cell data. *bioRxiv*, 2023.2001. 2008.523176 (2023).
- 6 Zheng, G. X. *et al.* Massively parallel digital transcriptional profiling of single cells. *Nature communications* **8**, 1-12 (2017).
- 7 Kocaoglu, M., Snyder, C., Dimakis, A. G. & Vishwanath, S. CausalGAN: Learning causal implicit generative models with adversarial training. *arXiv preprint arXiv:1709.02023* (2017).
- 8 Dahlin, J. S. *et al.* A single-cell hematopoietic landscape resolves 8 lineage trajectories and defects in Kit mutant mice. *Blood, The Journal of the American Society of Hematology* **131**, e1-e11 (2018).
- 9 Han, G. *et al.* Follicular lymphoma microenvironment characteristics associated with tumor cell mutations and MHC class II expression. *Blood cancer discovery* **3**, 428-443 (2022).
- 10 Lotfollahi, M., Wolf, F. A. & Theis, F. J. scGen predicts single-cell perturbation responses. *Nature methods* **16**, 715-721 (2019).

Reviewer #1 (Remarks to the Author):

The authors have addressed most of my concerns, and the revised paper is more valid and convincing. But I still have a few questions:

Major comments:

In the Introduction (lines 45-71), Results (lines 173-174), and Discussion (lines 588 - 612), the manuscript extensively described BoolODE and SERGIO, which also imposed GRNs as reference for scRNA-seq data simulation. However, the manuscript did not compare the performance of GRouNdGAN with them. I'm curious whether the two simulators could achieve comparable performance with GRouNdGAN if users finetuned their parameters appropriately.

In lines 232 to 246, the authors implemented TF knocked-out experiments to examine if the imposed GRN is effective in the simulated data. I'm confused about the sentence: "There was no change in the expression of genes that were not regulated by the knocked-out TF" (line 238). To my understanding, "the expression of genes that were not regulated by the knocked-out TF" will definitely not be changed, because GRouNdGAN only considered the edges between TFs and their genes. In addition, the expression of most target genes will be significantly changed after the knockout of their corresponding TFs, especially for those genes regulated by a single TF. This is because the input of the target generator only involved a random noise variable in such cases (without any regulatory information). It seems like a circular validation.

Minor comments:

There is a typo in the description of lines 3-4 of Table S6, Sheet 1: "Sheet3_BoneMarrow_GRNInference: Benchmark of different GRN inference methods on the PBMC-CTL dataset". "PBMC-CTL" should be revised as "BoneMarrow".

Reviewer #2 (Remarks to the Author):

I co-reviewed this manuscript with one of the reviewers who provided the listed reports as part of the Nature Communications initiative to facilitate training in peer review and appropriate recognition for co-reviewers.

Reviewer #2 (Remarks on code availability):

The authors provide detailed guide webpages to help users to employ their application.

Reviewer #3 (Remarks to the Author):

The authors provided a wide range of explanations and analysis to address my comments, and the authors have clarified that GRouNdGAN was developed for adding GRN information into existed scRNA-seq data rather than generating new scRNA-seq data based on GRN alone. I have two concerns about this point.

1. If GRouNdGAN is a reference-based scRNA generator, I suggest the authors provide more discussion to show GRouNdGAN's advancement compared to BEELINE, which can act as a reference-free scRNA generator (using an ODEs model to generate scRNA-seq data).
2. I suggest the authors provide some descriptions in Abstract and even Title about that GRouNdGAN requires existed scRNA-seq data as input. Current manuscript was somewhat misleading to readers that GRouNdGAN generates scRNA-seq data based on GRN alone.

Reviewer #1 (Remarks to the Author):

Comment: The authors have addressed most of my concerns, and the revised paper is more valid and convincing. But I still have a few questions:

Major comments:

Comment: In the Introduction (lines 45-71), Results (lines 173-174), and Discussion (lines 588 - 612), the manuscript extensively described BoolODE and SERGIO, which also imposed GRNs as reference for scRNA-seq data simulation. However, the manuscript did not compare the performance of GRouNdGAN with them. I'm curious whether the two simulators could achieve comparable performance with GRouNdGAN if users finetuned their parameters appropriately.

Response: It is important to note that, being a reference-based simulator, given a GRN, GRouNdGAN synthesizes realistic scRNA-seq data by aligning its learned distribution to that of a reference experimental single-cell dataset. In other words, GRN imposition and distribution matching to a reference are intertwined, and neither one is a post hoc step of the other.

BoolODE, unlike GRouNdGAN, is a reference-free (de novo) and rule-based simulator that relies on user-defined parameters instead of a reference dataset to artificially introduce properties of single-cell data (e.g., dropout, batch effect, existence of distinct populations, and variation among experimental samples and conditions). As a rule-based simulator, it allows for flexible selection of simulation parameters such as choice of noise strength, dropout rate, or kinetic parameters (for mRNA transcription, protein translation and mRNA and protein degradation rates). However, extensive user knowledge is required to leverage this flexibility to avoid generating unrealistic datasets (and recent studies have argued that one should use discretion when interpreting results obtained from such novo simulations¹). In fact, this was one of our motivations for developing GRouNdGAN.

These properties of BoolODE introduce several major challenges in comparing their performance against reference-based simulators (such as GRouNdGAN or other methods we benchmarked). First, selecting its set of parameters to make the BoolODE simulated data similar to a reference dataset is an extremely challenging task, since it is not designed for such a purpose. While one could fine-tune and select its parameters to generate simulated data that “generally” has characteristics of scRNA-seq data, aligning it to one particular reference dataset is quite challenging, if not impossible. In addition to the challenge of how one would even select the range of parameters to consider for fine-tuning BoolODE for this purpose, the bigger issue is that such fine-tuning process becomes user-dependent and subjective, since it is not automated and is not part of the BoolODE simulation pipeline. Moreover, besides the GRN structure, BoolODE requires a Boolean function defining the combinatorial effect of each gene’s TFs. Even if we make the simplifying assumption of Booleanity in gene co-regulation, we do not have access to such a GRN for datasets included in our study, presenting difficulties in running BoolODE with proper inputs. As a result of these challenges, we do not believe it is possible to fine-tune BoolODE to align with one particular reference dataset (and definitely not in a systematic and non-subjective manner), and as such, it cannot be evaluated using the same evaluation framework used in this study (since the evaluation framework used in this study are focused on how well the simulated data match one particular reference dataset,

instead of focusing on high-level distribution-related properties, such as number of zeros, library size, etc.).

The situation with SERGIO is slightly better with respect to the challenges above, since it is a hybrid *de novo*/reference-based simulator which employs a two-step procedure. In the first step, SERGIO simulates “clean” data following the GRN (a *de novo* process), supplemented by the cell-type-specific production rate of target genes and TFs. For TFs, the production rate is set to a constant and each target gene’s production rate is the sum of contribution it receives from its regulating TFs. Only in the second step is the reference dataset used to *manually* match the clean data to the reference data using technical noise, outlier genes, library size effects, and dropouts. In other words, a user must iteratively and manually vary several parameters and based on multiple metrics (described by authors of SERGIO), decide whether to halt the process or continue. Originally, we decided not to include SERGIO in our baselines, due to this manual, and more importantly, subjective process of deciding when the matching is complete. However, as requested by the reviewer, we decided to try this procedure and describe our experience in the revised version of the manuscript.

We used SERGIO to simulate cells from the PBMC-CTL dataset and employed the same GRN containing 15 regulating TFs per gene used to train GRouNdGAN. We computed the basal production rate b_i of each TF i from its mean expression x_i in the reference dataset, assuming a decay rate λ of 0.8:

$$b_i = \lambda E[x_i]$$

We followed SERGIO’s approach of uniformly sampling interaction strengths parameters K_{ij} (used to compute the production rate of target genes), denoting the maximum contribution of TF j to target gene i from a range of $[-1, -5] \cup [1, 5]$. Positive and negative contribution strengths K_{ij} represent activatory and repressive interactions, respectively.

Using the simulated clean dataset, we iteratively tuned the parameters of the following modules: outlier genes, dropouts, and conversion to UMI counts. But since all cells of the reference PBMC-CTL dataset were library-size normalized to a constant of 20000, we replaced SERGIO’s library size normalization module, which for each cell samples a library size from a lognormal distribution with one that assigns a library size of 20000 to all simulated cells. To compare the noise level between reference and simulated (now “noisy”) datasets, we used the same statistical measures as SERGIO (see Figure R1) and fine-tuned the parameters until we did not see any improvements from an iteration to the next. This process took around 2 hours. We provided the final technical noise parameters that we found in Table R1.

Table R1: Technical noise parameters found through iterative “clean” to reference PBMC-CTL dataset matching. Similar to reference data preprocessing, cells with less than 10 genes expressed in the simulated dataset were removed. Refer to the “METHOD DETAILS - Technical Noise” section of SERGIO for the description of technical noise module inputs and operation.

Outlier Genes			Dropouts		Library Size		Cell Filtering Threshold
π^o	μ^o	σ^o	k	q	μ^L	σ^L	
0.01	0.8	1	80	95	20000	0	10

Figure R1: SERGIO’s statistical parameters used to fine-tune its output to the PBMC-CTL reference dataset (related to Tables S4 and S5). “Clean” corresponds to the output of SERGIO before addition of technical noise, “Synth” refers to the dataset when technical noise was added to best match the reference dataset, and “Real” refers to the experimental PBMC-CTL reference dataset. The imposed GRN was the same GRN used with GRouNdGAN in Table 1. Panel A compares the library size distributions. Panels B and C show the distribution of zero counts per cell (normalized by number of genes) and per gene (normalized by number of cells), respectively. Panels D and E show the distribution of genes’ expression means and variances, respectively.

Tables R2 and R3 summarize the test and train-set performance of SERGIO before and after the described distribution matching. Overall, we observe an improvement in most metrics with the addition of technical noise to the “clean” dataset. However, even after distribution matching, SERGIO performs far worse than all simulators that we benchmarked. This is due to the fact that clean data is matched to the reference through five dataset-level (as opposed to gene-level or cell-level) statistics (as per procedure described by SERGIO’s authors). As such, the distribution of individual genes is not matched to the reference and gene identities are not preserved.

Table R2: Test set performance of SERGIO in generating realistic scRNA-seq data using the PBMC-CTL dataset. SERGIO (Clean) and SERGIO (Synth) refer to the simulated data before and after distribution matching through the addition of technical noise, respectively. The metrics are calculated between a

simulated dataset of 1000 cells and the held-out test set of 1000 real cells. For both SERGIO and GRouNdGAN, the same GRN was imposed. In this GRN, each gene was regulated by 15 TFs (constructed using GRNBoost2 from the experimental training set). For the first three metrics, a value closer to zero is preferred, for RF AUROC a value closer to 0.5 is preferred, and for miLISI a value closer to 2 is preferred. For the first two metrics, the values correspond to the distance of the mean centroids of the real and simulated cells. The RF AUROC of control corresponds to perfect performance (of a random classifier). The other control metrics are calculated using the two halves of the real test dataset.

Simulator	Cosine distance	Euclidean distance	MMD	RF AUROC	miLISI
SERGIO (Clean)	0.96998	5331	9.052	1.00	1.00
SERGIO (Synth)	0.86783	4913	6.247	1.00	1.00
GRouNdGAN	0.00057	182	0.026	0.54	1.89
Control	0.00019	99	0.012	0.50	1.91

Table R3: Training set performance of SERGIO in generating realistic scRNA-seq data using the PBMC-CTL dataset. SERGIO (Clean) and SERGIO (Synth) refer to the simulated data before and after distribution matching through the addition of technical noise, respectively. The metrics are calculated between a simulated dataset of 1000 cells and 1000 real cells randomly sampled from the training set. For both SERGIO and GRouNdGAN, the same GRN was imposed. In this GRN, each gene was regulated by 15 TFs (constructed using GRNBoost2 from the experimental training set). For the first three metrics, a value closer to zero is preferred, for RF AUROC a value closer to 0.5 is preferred, and for miLISI a value closer to 2 is preferred. For the first two metrics, the values correspond to the distance of the mean centroids of the real and simulated cells. The RF AUROC of control corresponds to perfect performance (of a random classifier). The other control metrics are calculated using the two halves of the real training dataset.

Simulator	Cosine distance	Euclidean distance	MMD	RF AUROC	miLISI
SERGIO (Clean)	0.96867	5248	9.026	1.00	1.00
SERGIO (Synth)	0.86998	5000	6.215	1.00	1.00
GRouNdGAN	0.00018	95	0.014	0.53	1.90
Control	0.00013	86	0.012	0.50	1.91

As can be seen from these tables, in spite of our efforts, the simulated data generated by SERGIO did not perform anywhere close to other simulators in our benchmarks. However, since the parameter tuning procedure of SERGIO relies on user decisions, we decided not to add them to the main tables of the study, but instead have included them in Supplementary File S1, describing it as our experience with SERGIO, instead of a definitive conclusion regarding its performance. A summary of these results was added to the main text (Page 11) and details were added to Supplementary Notes in File S1, Tables S4-S5, and Figure S4.

Comment: In lines 232 to 246, the authors implemented TF knocked-out experiments to examine if the imposed GRN is effective in the simulated data. I'm confused about the sentence: "There was no change in the expression of genes that were not regulated by the knocked-out TF" (line 238). To my understanding, "the expression of genes that were not regulated by the knocked-out TF" will definitely not be changed, because GRouNdGAN only considered the edges between TFs and their genes. In addition, the expression of most target genes will be significantly changed after the knockout of their corresponding TFs, especially for those genes regulated by a single TF. This is because the input of the target generator only involved a random noise variable in such cases (without any regulatory information). It seems like a circular validation.

Response: To respond to your comment, we have to break it down into two parts. The first part is related to your question regarding the lack of change in the expression of genes that were **not** regulated by the knocked-out TF. You are indeed correct that since such genes do not receive edges from such knocked-out TFs, their expression should not change. That is why the full sentence in this part reads as "There was no change in the expression of genes that were not regulated by the knocked-out TF **(as expected)**". The reason we have put "(as expected)" at the end of this sentence, is indeed to point out that this is not surprising and expected.

However, we have to respectfully disagree with regards to the second part of your comment that the observed significant changes of genes **regulated** by the knocked-out TFs is trivial. GRouNdGAN has two simultaneously important tasks: 1) generating realistic cells and 2) imposing the causal GRN. These two tasks, however, compete with each other and the model must find a balance between the two. Without the imposition of the GRN, it is relatively easy to generate realistic cells (as scGAN has done). Imposing the GRN without requiring the model to resemble a reference dataset is also relatively easy. However, achieving these two goals simultaneously is quite challenging. While developing GRouNdGAN, we had to make many different architectural changes and had to include auxiliary tasks such as labeler and anti-labeler to achieve this (our ablation study in Supplementary File S1 shows the effect of removing such aspects). In the earlier models that did not work well in imposing the GRN, the input to the target generators still were similar to GRouNdGAN's and included both a noise value and the TF expressions that regulate the target gene. However, it was easier for the model to simply use the noise input and ignore the TF inputs to generate realistic cells, achieving the first task of generating realistic cells, but not the second task of imposing the GRN.

So performing knockout experiments on the TFs is indeed needed, because it is quite possible (and we had seen that in earlier models) for the model to simply rely on noise and ignore the input TF values. We posit that the reason for the preference of the model to use noise, if additional auxiliary tasks are not used, is that the distribution of noise (aka Gaussian distribution) is much more flexible as an input than the expression of TFs that themselves were generated to mimic realistic TF expressions during pre-training.

We should also point out that we are not removing all TF inputs during the knockout experiments, but we perform this one TF at a time. So, for example, in the results reported for Figure 2F, each target gene generator receives input from one noise variable and 15 TFs. So setting these TFs to zero, one TF at a time, ensures that the effect of each *individual edge* is indeed imposed.

Finally, the KO experiment results are necessary to test whether what we intended to do during the training (aka imposing the GRN) has actually happened, the same way that even when we train the model and define its loss to ensure the samples are realistic, we still have to test whether the training procedure has achieved its goal by performing resemblance performance analysis.

We provided the following sentences to Pages 14-15 of the manuscript to emphasize some of the points above:

The analysis above shows that GRouNdGAN does not simply ignore the input TF values to rely solely on the noise input to generate realistic cells, and the effect of TF-gene edges are indeed imposed. Moreover, since each gene is regulated by multiple (15) TFs, but the knockout experiment is one TF at a time, this further shows that individual edges are imposed by the model.

Minor comments:

Comment: There is a typo in the description of lines 3-4 of Table S6, Sheet 1: "Sheet3_BoneMarrow_GRNInference: Benchmark of different GRN inference methods on the PBMC-CTL dataset". "PBMC-CTL" should be revised as "BoneMarrow".

Response: Thank you for spotting this type. We have now corrected that.

Reviewer #2 (Remarks to the Author):

Comment: I co-reviewed this manuscript with one of the reviewers who provided the listed reports as part of the Nature Communications initiative to facilitate training in peer review and appropriate recognition for co-reviewers.

Reviewer #2 (Remarks on code availability):

Comment: The authors provide detailed guide webpages to help users to employ their application.

Reviewer #3 (Remarks to the Author):

Comment: The authors provided a wide range of explanations and analysis to address my comments, and the authors have clarified that GRouNdGAN was developed for adding GRN information into existed scRNA-seq data rather than generating new scRNA-seq data based on GRN alone. I have two concerns about this point.

1. If GRouNdGAN is a reference-based scRNA generator, I suggest the authors provide more discussion to show GRouNdGAN's advancement compared to BEELINE, which can act as a reference-free scRNA generator (using an ODEs model to generate scRNA-seq data).]

Response: Thank you for this suggestion. We updated the Discussion on pages 32-34 to clarify the differences and benefits of GRouNdGAN compared to BoolODE, the simulator used in BEELINE, and also SERGIO.

Comment: 2. I suggest the authors provide some descriptions in Abstract and even Title about that GRouNdGAN requires existed scRNA-seq data as input. Current manuscript was somewhat misleading to readers that GRouNdGAN generates scRNA-seq data based on GRN alone.

Response: Thank you for this suggestion. It was challenging to change the title, due to it becoming too long, but we modified the abstract and introduction to clarify this point early on.

We modified the following part of the abstract (parts that point to the input reference dataset are underlined):

“We introduce GRouNdGAN, a gene regulatory network (GRN)-guided reference-based causal implicit generative model for simulating single-cell RNA-seq data, *in-silico* perturbation experiments, and benchmarking GRN inference methods. Through the imposition of a user-defined GRN in its architecture, and using a reference scRNA-seq dataset, GRouNdGAN simulates steady-state and transient-state single-cell datasets where genes are causally expressed under the control of their regulating transcription factors (TFs). Training on six experimental reference datasets (from four independent studies), ...”

We modified the following part of introduction (parts that point to the input reference dataset are underlined):

“GRouNdGAN (GRN-guided *in silico* simulation of single-cell RNA-seq data using Causal generative adversarial networks) is a causal implicit generative model for reference-based GRN-guided simulation of scRNA-seq data inspired by CausalGAN. Given an input GRN and a reference scRNA-seq dataset, it can be trained to generate simulated data that is both indistinguishable from the reference data and faithful to the causal regulatory interactions of the input GRN.”

References

- 1 Crowell, H. L., Morillo Leonardo, S. X., Soneson, C. & Robinson, M. D. The shaky foundations of simulating single-cell RNA sequencing data. *Genome Biology* **24**, 1-19 (2023).

Reviewer #1 (Remarks to the Author):

All my concerns have been addressed and I have no further comments.

Reviewer #2 (Remarks to the Author):

I co-reviewed this manuscript with one of the reviewers who provided the listed reports as part of the Nature Communications initiative to facilitate training in peer review and appropriate recognition for co-reviewers.

Reviewer #3 (Remarks to the Author):

My concerns have been addressed and I think the manuscript is ready to be published.